# The ABA/LANCL1/2 Hormone/Receptor System Controls Adipocyte Browning and Energy Expenditure

**DOI:** 10.3390/ijms24043489

**Published:** 2023-02-09

**Authors:** Sonia Spinelli, Vanessa Cossu, Mario Passalacqua, Jacob B. Hansen, Lucrezia Guida, Mirko Magnone, Gianmario Sambuceti, Cecilia Marini, Laura Sturla, Elena Zocchi

**Affiliations:** 1Section of Biochemistry, Department of Experimental Medicine, University of Genova, Viale Benedetto XV 1, 16132 Genova, Italy; 2IRCCS Ospedale Policlinico San Martino, U.O. Medicina Nucleare, 16132 Genova, Italy; 3Department of Biology, University of Copenhagen, DK-2100 Copenhagen, Denmark; 4Department of Health Sciences, University of Genoa, 16132 Genova, Italy; 5Institute of Molecular Bioimaging and Physiology (IBFM), National Research Council (CNR), 20054 Milan, Italy

**Keywords:** AMPK/PGC-1α/Sirt1 signaling axis, ERRα, glucose transport and oxidation, mitochondrial biogenesis and respiration, OXPHOS uncoupling, β-adrenergic receptor, thyroid receptors, DIO2, UCP1/3, energy metabolism, thermogenesis

## Abstract

The abscisic acid (ABA)/LANC-like protein 1/2 (LANCL1/2) hormone/receptor system regulates glucose uptake and oxidation, mitochondrial respiration, and proton gradient dissipation in myocytes. Oral ABA increases glucose uptake and the transcription of adipocyte browning-related genes in rodent brown adipose tissue (BAT). The aim of this study was to investigate the role of the ABA/LANCL system in human white and brown adipocyte thermogenesis. Immortalized human white and brown preadipocytes, virally infected to overexpress or silence LANCL1/2, were differentiated in vitro with or without ABA, and transcriptional and metabolic targets critical for thermogenesis were explored. The overexpression of LANCL1/2 increases, and their combined silencing conversely reduces mitochondrial number, basal, and maximal respiration rates; proton gradient dissipation; and the transcription of uncoupling genes and of receptors for thyroid and adrenergic hormones, both in brown and in white adipocytes. The transcriptional enhancement of receptors for browning hormones also occurs in BAT from ABA-treated mice, lacking LANCL2 but overexpressing LANCL1. The signaling pathway downstream of the ABA/LANCL system includes AMPK, PGC-1α, Sirt1, and the transcription factor ERRα. The ABA/LANCL system controls human brown and “beige” adipocyte thermogenesis, acting upstream of a key signaling pathway regulating energy metabolism, mitochondrial function, and thermogenesis.

## 1. Introduction

The appearance of ABA dates back to unicellular algae, and the hormone is conserved across kingdoms in modern higher plants and mammals [1]. In plants, the pleiotropic functions of ABA are summarized in its definition as a “stress hormone”, i.e., a hormone that triggers plant responses to variations in environmental conditions. Indeed, ABA orchestrates multiple, tissue-specific responses in plants to biotic as well as abiotic “stressors”, the latter including water, light, and nutrients availability. In lower Metazoa, a similar role for ABA has been observed in sponges, where changes in water temperature induces release of ABA, which stimulates filtration and respiration [2].

In mammals, environmental stimuli reaching cells are mostly (bio)chemical rather than physical and they reach cells via the bloodstream. Still, a relic of this evolutionary past can be traced in the role of ABA in nitric oxide generation by UVB-stimulated human keratinocytes [3], similarly to what occurs in higher plants [4].

In humans, plasma ABA increases after a glucose load [5] and oral ABA reduces glycemia, while at the same time reducing plasma insulin [6]. These results indicated a role for ABA in mammalian glycemia control and spurred research aimed at understanding the molecular mechanisms underlying this general effect of ABA on glucose metabolism.

Skeletal muscle and adipose tissue are primarily responsible for whole body glucose consumption, partly due to their significant combined mass and tissue-specific metabolism: skeletal muscle consumes large quantities of ATP for contraction, and white and brown adipocytes are proficient in triglyceride synthesis from glucose and in energy-consuming thermogenesis, respectively. Thus, skeletal muscle and adipose tissue were the obvious targets of investigations aimed at understanding the mechanisms through which ABA improves whole body glucose disposal.

The mammalian ABA receptors identified so far belong to the LANCL family, which dates back to bacteria, where homologs of these proteins are responsible for the synthesis of cyclic peptides with antibiotic properties. A comprehensive study demonstrated that mammalian LANCL proteins do not synthesize lanthipeptides [7]. Indeed, solid evidence obtained on recombinant human LANCL1 and LANCL2 indicates that they both bind ABA with a submicromolar affinity [8,9]. The identification of these ABA receptors allowed their overexpression or silencing in murine skeletal myocytes and in adipocytes to study the effect of this genetic manipulation on glucose metabolism.

The ABA/LANCL1-2 hormone/receptor system indeed plays an important role in skeletal muscle energy metabolism by stimulating transcriptional and metabolic processes, leading to increased glucose oxidation and mitochondrial function. Contrary to insulin, which activates Akt, ABA relies on an AMP-activated protein kinase (AMPK)/peroxisome proliferator-activated receptor γ co-activator 1α (PGC-1α)/sirtuin 1 (Sirt1)-dependent signaling pathway to stimulate glucose transport and metabolism in muscle cells [9,10]. Therefore, the ABA/LANCL1/2 system provides an insulin-independent alternative pathway, capable of stimulating muscle glucose disposal and energy metabolism. Indeed, the administration of ABA improves the effect of exogenous low-dose insulin or of residual endogenous insulin in murine models of insulin-dependent type 1 diabetes [11]. Through the master transcriptional regulator PGC-1α, the ABA/LANCL1-2 system stimulates key mitochondrial functions in muscle cells, including biogenesis, respiration, transcription of uncoupling proteins (UCPs) and sarcolipin, NAD^+^ synthesis via NAMPT, and the mitochondrial proton gradient [9]. Skeletal muscle plays a pivotal role in whole body glucose disposal: a reduction of muscle glucose metabolism, as occurs in diabetes, results in glucose intolerance, conversely a stimulation of muscle glucose uptake and consumption, as occurs after insulin administration or during exercise, improves glucose tolerance. Indeed, prediabetic or borderline subjects show a significant amelioration of fasting glycemia, glucose tolerance and glycated hemoglobin after chronic treatment with ABA [12,13].

In addition to skeletal muscle, adipose tissue is another major contributor to whole body energy balance directly via white adipose tissue (WAT) triglyceride turnover and brown adipose tissue (BAT) energy expenditure for thermogenesis and indirectly via the release of numerous adipokines and batokines, regulating body metabolism, tissue sensitivity to other hormones, and food intake. Both LANCL 1 and LANCL2 are expressed in adipose tissue (Appendix A). In line with what was observed in skeletal muscle, a role for the ABA/LANCL system in WAT/BAT physiology was also reported. Several in vitro and in vivo results indicate that ABA stimulates WAT browning and BAT activity in rodents.

Nanomolar ABA, via LANCL2, stimulates glucose transporter 4 (GLUT4) expression and glucose uptake in differentiated murine 3T3-derived (white) adipocytes. At variance with insulin, however, ABA does not per se induce adipocyte differentiation; when added during insulin-induced adipocyte differentiation process, ABA instead promotes cell remodeling with a reduction of cell size and triglyceride content. Interestingly, LANCL2 silencing abrogates and LANCL2 overexpression instead enhances the effect not only of ABA, but also of insulin on adipocyte glucose uptake. The treatment of differentiated 3T3-derived white adipocytes with ABA increases the mitochondrial respiration and transcription of adiponectin, leptin, and of several browning genes, including UCP1. Finally, chronic ABA treatment increases the mitochondrial content and transcription of browning genes, including UCP1, in WAT and BAT of CD1 mice, and a single oral dose of ABA stimulates BAT glucose uptake in rats [14]. Altogether, these data indicate that the activation of the ABA/LANCL2 system promotes the browning of murine white adipocytes and BAT activity in rodents, suggesting their role in the control of whole-body energy balance; indeed, CD1 mice fed a high glucose diet containing ABA show a reduced body weight gain as compared with the controls, which were fed the same diet without the addition of ABA [10].

A possible role for LANCL1 in the ABA-mediated effects on adipocytes was not explored in these studies; however, it is likely that LANCL1 could participate in the ABA signaling pathway in adipocytes, similarly to what observed in skeletal myocytes [9].

Taken together, results obtained on rodents, both in vitro and in vivo, point to a regulatory role of the ABA/LANCL hormone/receptor system in glucose disposal for energy metabolism by activating glucose uptake and oxidation, mitochondrial biogenesis and respiration, and the transcription of uncoupling proteins in both muscle and adipose tissue, thus increasing metabolic energy dissipation.

The combined effect of an ABA-induced, insulin-independent, increased muscle and adipocyte glucose uptake, oxidation, and metabolic energy dissipation may explain the observed reduction of body weight in chronically ABA-treated mice and in subjects with borderline glucose tolerance [12].

Interventions aimed at increasing whole-body energy consumption lie at the heart of non-surgical therapeutic strategies to reduce body weight. The pharmacological induction of WAT browning, BAT activation, and stimulation of WAT/muscle glucose uptake are among the currently investigated treatments, not devoid of possible side-effects due to the systemic effects of the hormones/agonists/drugs employed to this end, e.g., thyroid hormone analogs, β-adrenergic receptor agonists, and glitazones. Although studies on rodents provide valuable preclinical information, abundance, body localization, and physiology of white and brown adipose tissue are not exactly comparable in rodents and humans [15]; for this reason, studies on rodents, though highly informative, particularly when performed on genetically modified strains, are not immediately transferable to the clinical setting.

The aim of this study was to investigate the role of the ABA/LANCL1-2 system in the differentiation of human white and brown preadipocytes. Human LANCL1 and LANCL2 were overexpressed or silenced in immortalized white or brown human preadipocytes, and the transcriptional and metabolic effects induced by these genetic manipulations were studied during in vitro differentiation. In addition, the effect of chronic ABA treatment on WAT and BAT transcriptional levels of browning genes and on mitochondrial DNA content was explored in LANCL2 knock-out (KO) mice.

## 2. Results

### 2.1. Transcriptional Effects of ABA on Human Brown and White Adipocytes

Previous studies on the effect of ABA on adipocyte and muscle cells showed that ABA, via LANCL2, stimulates the transcription of several browning genes in murine 3T3-L1 preadipocytes [14] and that ABA, via LANCL1 and LANCL2, activates the AMPK-1α /PGC-1α/Sirt1 signaling axis in muscle cells, thereby stimulating glucose uptake and oxidation and mitochondrial biogenesis, oxygen consumption, and proton gradient maintenance [9,16]. In adipocytes, the same signaling axis controls the browning process of white adipocytes and stimulates BAT activity [14]. In order to study a possible effect of ABA on human white and brown adipocyte differentiation, immortalized white (TERT-hWA) and brown (TERT-hBA) human preadipocytes were induced to differentiate with a white- or brown-specific differentiation cocktail (see Appendix A), with or without (controls) 100 nM ABA. At the end of white differentiation (day 12), white adipocytes were further incubated for 3 days with 1 μM rosiglitazone to induce browning.

At microscopic examination, approximately 90% of white and brown adipocytes became lipid-laden at the end of differentiation (day 12), with larger and more densely packed vesicles in TERT-hWA (Figure 1A). At the end of the “browning” process of white adipocytes (day 15), lipid droplets were visibly smaller, as occurs in BAT and in line with previous observations [17] (Figure 1A, lower panels). In the absence of a differentiation-inducing cocktail, ABA alone did not induce any accumulation of lipid droplets (not shown), as already observed in murine 3T3-L1 preadipocytes [14]. Moreover, the presence of ABA during differentiation did not induce relevant changes in the microscopic features of differentiated white or brown adipocytes.

The transcription of the mRNAs related to the AMPK/PGC-1α/Sirt1 axis and to selected browning genes was explored on days 0 and 12 during brown differentiation of TERT-hBA cells and white differentiation of TERT-hWA and also at day 15 in mature white adipocytes exposed to a “browning” cocktail. The transcription of AMPK, PGC-1α, and Sirt1 increased during differentiation as compared with day 0, both in white and brown adipocytes, and the presence of ABA further increased their mRNA levels on day 12 (Figure 1B, upper panel). In white adipocytes, the treatment with either 100 nM ABA or 1 µM rosiglitazone (a known browning agent) to the basal medium-containing differentiation cocktail (days 12–15) further increased the transcription of the AMPK/PGC-1α/Sirt1 axis, and the combination of rosiglitazone and ABA induced an even greater transcriptional effect (Figure 1B, upper panel).

The mRNA levels of the browning-specific markers UCP1, CIDE-A, and PPAR-γ [18,19,20,21] and of GLUT4 were up-regulated after brown adipocyte differentiation of TERT-hBA (day 12) and also after the “browning” of mature white adipocytes derived from TERT-hWA (day 15) (Figure 1B, center left panel). When present during brown adipocyte differentiation, ABA significantly increased the transcription of UCP1, CIDE-A, and GLUT4 at day 12. The presence of either 100 nM ABA or 1 µM rosiglitazone similarly increased the expression of browning genes in white adipocytes while their combination did not induce a greater effect. In “beige” white adipocytes, the transcription of UCP1 and CIDE-A was significantly higher in rosiglitazone- vs. ABA-treated cells while the opposite occurred for GLUT4, suggesting that the effect of ABA does not occur via the same signaling pathway as rosiglitazone. Mitochondrial DNA increases by 50% during brown, but not white, adipocyte differentiation; the presence of ABA during brown differentiation of TERT-hBA increased mitochondriogenesis and the highest levels were again observed in mature white adipocytes treated with rosiglitazone and ABA together during the “browning” process, again suggesting non-overlapping signaling pathways (day 15) (Figure 1B, center right panel). The expression levels of enzymes involved in oxidative metabolism were also investigated. The transcription of CPT1β (carnitine palmitoyltransferase 1 beta, necessary for fatty acid transport into mitochondria), MPC1 (mitochondrial pyruvate carrier 1, necessary for mitochondrial pyruvate transport), and PDHα1 (pyruvate dehydrogenase subunit alpha 1, necessary for pyruvate oxidation to acetyl-CoA) were all upregulated during adipocyte, particularly white, differentiation (Figure 1B, lower left panel). The presence of ABA slightly increased transcription of these genes, both during brown and white adipocyte differentiation (day 12) while rosiglitazone particularly increased the transcription of CPT1β in “beige” white adipocytes, suggesting a higher fatty acid oxidation rate in rosiglitazone vs. ABA-treated white adipocytes after browning. Finally, the transcription of the three LANCL proteins was also investigated during human adipocyte differentiation. LANCL1 and LANCL2, but not LANCL3 mRNA levels, increased significantly during both white and brown pre-adipocyte differentiation, particularly in the presence of ABA or of rosiglitazone (Figure 1B, lower right panel), suggesting that LANCL1 and LANCL2 could play a role in human preadipocyte differentiation and also in the effect of rosiglitazone.

Taken together, these results demonstrate that, similarly to what observed in murine preadipocytes, ABA alone does not induce adipocyte differentiation in human preadipocytes; however, when added to a white- or brown-inducing differentiation cocktail, ABA substantially increases transcription of the AMPK/PGC-1α/Sirt1 axis, which in turn controls both white and brown adipocyte metabolism. Both ABA and rosiglitazone stimulate the transcription of browning genes in white adipocytes, indicating a similar “browning” effect on human preadipocytes; at variance with rosiglitazone, ABA has a more pronounced stimulatory effect on GLUT4 transcription and on mitochondrial DNA content.

### 2.2. The ABA/LANCL1/2 Hormone/Receptor System Controls Protein Levels of AMPK, pAMPK, and PGC-1α in Human Brown and White Preadipocytes

The role of LANCL1 and LANCL2 in human adipocyte differentiation was further explored by overexpressing these proteins in brown and white preadipocytes (TERT-hBA and TERT-hWA, respectively). The overexpression of LANCL1 and LANCL2 was induced by retroviral infection and was confirmed by Western blot analysis.

All experiments were performed on brown (Figure 2A, upper panels) and on white preadipocytes (Figure 2B, upper panels) expressing approximately 6- and 15-times higher levels of LANCL1 and LANCL2, respectively, then control cells, infected with the empty PLV vector. Interestingly, a small but significant increase of LANCL1 in LANCL2-overexpressing cells, and vice versa, was observed both in brown and white undifferentiated preadipocytes, suggesting the existence of a transcriptional link between these proteins, as already observed in skeletal muscle [9] and in cardiomyocytes [16].

The overexpression of LANCL1, LANCL2, or of both proteins together in human brown preadipocytes (TERT-hBA) did not significantly modify protein levels of AMPK, pAMPK, or PGC-1α as compared with PLV-infected control cells (Figure 2A, central and lower panels, grey bars). However, treatment with 100 nM ABA for 1 h induced an approximately 2-fold increase in AMPK phosphorylation (pAMPK) and PGC-1α protein levels in cells overexpressing LANCL1 or LANCL2 relative to untreated cells. pAMPK increased more than total AMPK protein expression upon ABA treatment of the cells, leading to a doubling of the pAMPK/AMPK ratio (Figure 2A, central right panel).

In undifferentiated white adipocytes (TERT-hWA) significantly higher protein levels of AMPK, pAMPK, and PGC-1α were observed in LANCL1/2-overexpressing cells relative to PLV-infected cells and treatment with ABA further significantly increased protein levels of the AMPK, pAMPK, and PGC-1α axis (Figure 2B, central and lower panels). In ABA-treated white and brown preadipocytes alike, the combined LANCL1/2 overexpression did not induce a higher increase of pAMPK or of PGC-1α protein than observed in cells overexpressing either LANCL1 or LANCL2, indicating a redundancy of receptor function, as also observed in muscle cells [9].

The increase of both total and of phosphorylated (active) AMPK observed in LANCL1/2-overexpressing human brown and white preadipocytes necessarily results in a higher catalytic activity.

The AMPK/PGC-1α/Sirt1 axis was investigated at the protein level also in human white and brown preadipocytes silenced for the expression of LANCL1/2. The efficiency of shRNA-mediated silencing was similar for LANCL1 and LANCL2 and resulted in an approximately 80% reduction of both mRNA and protein levels in human brown (Figure 3A, upper panels) and white preadipocytes (Figure 3B, upper panels). The results obtained were specular to those described for overexpressing cells (Figure 2). The silencing of LANCL1/2 proteins resulted in a significant (at least 50%) reduction in the protein levels of AMPK, pAMPK, and PGC-1α, both in brown (Figure 3A, central and lower panels) and in white preadipocytes (Figure 3B, central and lower panels). In addition, in LANCL1/2-silenced cells ABA did not increase AMPK, pAMPK, or PGC-1α protein levels above those measured in ABA-treated control cells infected with a scrambled (SCR) shRNA, nor did it modify the pAMPK/AMPK ratio.

Altogether, the results obtained on human brown and white undifferentiated preadipocytes demonstrate that the ABA/LANCL1/2 system positively controls the AMPK, pAMPK, or PGC-1α axis at the transcriptional and protein level.

### 2.3. The ABA/LANCL1/2 Hormone/Receptor System Controls Browning and Mitochondrial Gene Expression in Differentiated White and Brown Human Adipocytes

Next, we investigated the effect of LANCL1/2 overexpression on transcription of the AMPK/PGC-1α/Sirt1 axis during white or brown adipocyte differentiation.

As shown in Figure 4A, the transcription of these genes increased significantly in differentiated LANCL1/2-overexpressing white (right panel), brown (left panel), and white “beige” adipocytes (right panel) as compared with control cells infected with the empty vector (PLV) and treatment with ABA further increased transcription. The AMPK/PGC-1α/Sirt1 axis controls energy metabolism and mitochondrial respiration in myocytes and adipocytes [9,10,14,16,22,23], suggesting to investigate possible metabolic consequences of the activation of this pathway by LANCL1/2 overexpression.

GLUT4 transcription increased significantly (4- and 2-fold in brown and white adipocytes, respectively, relative to control cells infected with the empty vector PLV) in differentiated (day 12) LANCL1/2-overexpressing adipocytes. The treatment with ABA induced a further increase in GLUT4 transcription, which was similarly 4- and 2-fold higher in LANCL1/2-overexpressing cells than in cells infected with the empty vector (Figure 4B,C, upper central panels).

The transcription of key mitochondrial transporters/enzymes involved in oxidative metabolism (MPC1, CPT1β and PDHα1) was markedly upregulated in LANCL1/2-overexpressing human white and brown adipocytes during differentiation, as compared with PLV-infected controls and further increased in cells treated with 100 nM ABA during differentiation (Figure 4B,C, upper right panels).

Mitochondrial DNA, a proxy measurement for mitochondrial number, increased up to twice the Day 0 levels during brown or white adipocyte differentiation and only slightly more in LANCL1/2-overexpressing, differentiated, ABA-treated white and brown adipocytes (Figure 4B,C, lower right panels). Transcription of mitochondrial NADH:ubiquinone oxidoreductase core subunit 1 (MT-ND1) instead increased logarithmically (approximately 15-times) in LANCL1/2-overexpressing, differentiated adipocytes, both white and brown, particularly when treated with ABA during differentiation (Figure 4B,C, lower right panels). This unexpected result suggested that in LANCL1/2-overexpressing adipocytes mitochondria were not so much increased in number as being more proficient in their respiratory function.

The “browning” of differentiated white adipocytes, induced by the addition of rosiglitazone on days 12–15 of culture (see Appendix A), resulted in a 5-fold increase of UCP1 transcription in PLV-infected cells (day 15), relative to mature white adipocytes (day 12) and treatment with ABA during browning tripled the increase in mRNA levels of this uncoupling protein (Figure 4C, upper right panel). In white adipocytes overexpressing LANCL1/2 the increase in UCP1 mRNA was exponential relative to PLV-infected controls, even in the absence of rosiglitazone-mediated induction, and further increased after treatment with ABA (Figure 4C, upper right panel), indicating a role for the ABA/LANCL1/2 system in the control of UCP1 transcription during browning of human white adipocytes. UCP3 also significantly increased in white adipocytes overexpressing LANCL1/2, particularly after treatment with ABA (Figure 4C, upper right panel). Finally, the transcription of the ADP/ATP translocator ANT1, which participates in proton gradient dissipation together with UCP1/3 [23,24], was also significantly higher in LANCL1/2-overexpressing brown (Figure 4B, upper right panel) and white adipocytes (Figure 4C, upper right panel) relative to their respective controls.

In addition to enhancing transcription of uncoupling proteins, overexpression of LANCL1/2 increased mRNA levels of several key hormone receptors and enzymes responsible for white adipocyte “browning” and brown adipocyte activation; indeed, similar results were observed on brown (Figure 4B, lower left panel) and white human preadipocytes (Figure 4C, lower left panel). The transcription of β3-adrenergic receptor (ADRβ3), thyroid hormone receptor α1 and β (THRα1, THRβ), insulin receptor (INSR), estrogen-related receptor α (ERRα), and the enzyme deiodinase 2 (DIO2), necessary for T4-to-T3 conversion inside target cells, all increased significantly more in LANCL1/2-overexpressing adipocytes compared with controls. The increase of ERRα in LANCL1/2-overexpressing, ABA-treated adipocytes, both white and brown, was particularly impressive (20-fold).

The mRNA levels of LANCL1/2 proteins increased during white or brown adipocyte differentiation and also during “browning” of white adipocytes (Figure 1B, lower right panel). Surprisingly, the transcription of LANCL1 and LANCL2 also significantly increased during differentiation of LANCL1/2-overexpressing cells (Figure 4B,C, central lower panels). As the transcription of retrovirally-transduced inserts should be independent of transcriptional regulatory mechanisms, the higher mRNA levels of the LANCL1/2 proteins should reflect increased expression of the endogenous proteins.

The same transcriptional targets explored in LANCL1/2-overexpressing cells (Figure 4) were also analyzed in human white and brown adipocytes silenced for the expression of LANCL1/2 (Figure 5). The results obtained were opposite to those observed in overexpressing cells. The silencing of LANCL1/2 proteins resulted in a significant (at least 50%) reduction of the mRNA levels of all transcriptional targets previously shown to be upregulated in LANCL1/2-overexpressing cells, both in brown (Figure 5A left panel and Figure 5B) and in white adipocytes (Figure 5A right panel and Figure 5C).

### 2.4. Overexpression of LANCL1 and LANCL2 Increases, and Their Combined Silencing Reduces, Mitochondrial Uncoupling in TERT-hBA and TERT-hWA Adipocytes

To evaluate whether TERT-hBA and TERT-hWA adipocytes overexpressing LANCL1 and LANCL2 treated with 100 nM ABA or T3 possess thermogenic features, respiration measurements were performed to assess their mitochondrial function using the Seahorse XFp Analyzer, with the sequential addition of oligomycin (an inhibitor of the ATP synthase proton channel, which allows to calculate ATP-linked respiration), FCCP (a proton transporter, which completely dissipates the mitochondrial proton gradient, allowing the calculation of the maximal respiration rate) and rotenone/antimycin A (inhibitors of the electron transfer chain, which completely inhibit mitochondrial respiration, allowing the measurement of baseline, non-mitochondrial oxygen consumption). The partial dissipation of the mitochondrial proton gradient (∆Ψ), e.g., under the action of the uncoupling hormone T3, is expected to reduce the increase of cell oxygen consumption upon addition of FCCP (maximal respiration), since ∆Ψ is already dissipated, and also to reduce ATP-linked respiration since most of the ∆Ψ is consumed by heat production instead of by ATP synthesis.

Significantly higher maximal and ATP-linked respiration rates were observed in brown adipocytes overexpressing LANCL1/2 compared with control cells, infected with the empty vector PLV (Figure 6A, upper panels). In addition, a significantly increased gap between maximal respiration with or without T3 (Figure 6A, left upper panel), and with or without ABA (Figure 6A, right upper panel), was observed in LANCL1/2-overexpressing cells compared with controls, indicating that mitochondrial respiration was already uncoupled in T3- and ABA-treated brown adipocytes cells, before the addition of FCCP. Indeed, the reduction of ATP-linked respiration in the presence vs. absence of T3 or ABA was more evident in LANCL1/2-overexpressing brown adipocytes as compared with control (PLV) cells (Figure 6A, upper panels). The combined silencing of LANCL1/2 in brown adipocytes conversely resulted in a significant reduction of basal and maximal respiration rates compared with control cells (Figure 6A, lower panels). Moreover, maximal respiration in the presence of either T3 or ABA was not reduced compared with that without hormones, indicating that the silencing of LANCL1/2 did not allow the uncoupling effect of T3 (Figure 6A, left lower panel) and of ABA (Figure 6A, lower right panel) to occur (Figure 6A, lower panels).

The overexpression of LANCL1/2 in white adipocytes did not result in a significant increase of basal and maximal respiration rates (Figure 6B, upper panels), as observed in brown adipocytes. Maximal and ATP-linked respiration rates were similar in T3-treated and untreated LANCL1/2-overexpressing cells, as also observed in PLV-infected control cells, indicating that mature white adipocytes do not respond to T3 with a mitochondrial uncoupling, as indeed expected (Figure 6B, upper left panel). Conversely, treatment with ABA reduced both maximal and ATP-linked respiration in white adipocytes overexpressing LANCL1/2, indicating an uncoupling effect of ABA on white adipocytes overexpressing LANCL proteins (Figure 6B, upper right panel). The combined silencing of LANCL1/2 significantly reduced both basal and maximal respiration also in white adipocytes, as observed in brown adipocytes (Figure 6B, lower panels). In addition, LANCL1/2 silencing also abrogated the uncoupling effect of ABA, as maximal and ATP-dependent respiration rates were similar in ABA-treated vs. -untreated cells (Figure 6B, lower right panel).

Uncoupled respiration was also calculated as the difference between the oligomycin and the rotenone/antimycin A injections (Appendix A). These data confirm an increased mitochondrial uncoupling in LANCL1/2 overexpressing cells and conversely a reduction in LANCL1/2-double silenced cells, compared with their respective controls. Moreover, these new calculations confirm that the uncoupling effect of ABA on “beige” adipocytes is higher than that of T3.

Altogether, these results indicate that in LANCL1/2-overexpressing brown adipocytes mitochondrial uncoupling is increased and the difference between maximal and ATP-linked respiration is further increased in the presence of ABA. A similar, though less evident uncoupling occurs in LANCL1/2-overexpressing “beige” adipocytes, where the effect of ABA exceeds that of T3.

### 2.5. Mitochondrial Function and Proton Gradient Increase in LANCL1/2-Overexpressing and Are Conversely Reduced in LANCL1/2-Silenced White and Brown Adipocytes

The significant, and opposite, modification of basal and maximal respiration rates observed in LANCL1/2-overexpressing vs. double silenced adipocytes suggested to directly compare the mitochondrial proton gradient in these cells. LANCL1/2-overexpressing or -silenced adipocytes were stained with MitoTracker Deep Red FM dye, which passively diffuses across the plasma membrane and accumulates into metabolically active mitochondria.

Both white and brown adipocytes overexpressing LANCL1/2 showed an approximate 4-fold increase of mitochondrial fluorescence compared with the respective controls, infected with the empty vector (Figure 7A, panels A, E, C and G). The treatment with ABA further increased mitochondrial fluorescence to approximately 6-fold higher levels compared with control (PLV-infected), ABA-untreated cells (Figure 7A, panels B, F, D, and H). In LANCL1/2 double-silenced white and brown adipocytes, conversely, the mitochondrial number was reduced by approximately 50% compared with control cells and treatment with ABA was without effect (Figure 7B). The MitoTracker fluorescent dye enters into mitochondria thanks to their membrane potential; thus, a steeper mitochondrial proton gradient (∆Ψ) could affect the extent of dye accumulation within mitochondria.

To directly investigate the magnitude of the ∆Ψ in LANCL1/2-overexpressing or -silenced adipocytes, we used the ∆Ψ-sensitive dye JC-1. This fluorescent molecule accumulates within mitochondria and changes its emission from green to red as the ∆Ψ increases [25].

As shown in Figure 8A, mitochondrial fluorescence was largely red in LANCL1/2-overexpressing brown or white adipocytes (rows C and G) and approximately 6-fold higher than that measured in control cells, infected with the empty vector. The red fluorescence further significantly increased in ABA-treated cells (rows D and H). Conversely, mitochondrial fluorescence was predominantly green in LANCL1/2-silenced cells (Figure 8B, rows C and G), and red fluorescence was almost undetectable even after addition of ABA (Figure 8B, rows D and H). The calculated red/green ratio (Figure 8) was approximately 7-fold higher in overexpressing cells and 5-fold lower in double silenced cells, compared with the respective control. Adipocytes transformed with the two different control vectors (PLV for the overexpression and shRNA-SCR for the silencing of the LANCL proteins) had similar red/green ratios, and the calculated values were in between those of the overexpressing and the double-silenced adipocytes. Thus, the marked difference in the mitochondrial ∆Ψ between overexpressing and silenced cells was not attributable to the different viral vectors, but to the overexpression vs. silencing of the LANCL1/2 proteins.

Collectively, these results indicate that LANCL1/2-overexpression significantly increases while the combined LANCL1/2-silencing reduces, mitochondrial function, and ∆Ψ both in brown and white human adipocytes. The effect is greatest on white adipocytes induced to brown. Moreover, exogenous ABA increases the mitochondrial proton gradient in the overexpressing cells, and this effect is almost non-existing in the double-silenced cells.

### 2.6. The Combined LANCL1/2 Overexpression Increases, and Their Silencing Reduces, Glucose Uptake in Differentiated Brown and White Human Adipocytes

Mitochondrial respiration depends on nutrient oxidation and coenzyme reduction. Thus, an increased respiration rate must be sustained with increased nutrient availability. LANCL1/2 overexpression or silencing stimulates and reduces, respectively, GLUT4 transcription in terminally differentiated white and brown adipocytes (Figure 4B,C).

Glucose uptake of LANCL1/2-overexpressing and -silenced adipocytes was directly measured by means of the fluorescent and metabolically inert glucose analog 2-NBDG (Figure 9A). The overexpression of the LANCL proteins increased 2-NBDG uptake 2-fold (left panel) and 5-fold (right panel) in brown and white adipocytes, respectively, relative to their controls, infected with the empty vector PLV. Conversely, NBDG uptake was greatly reduced in LANCL1/2-double silenced adipocytes, both brown (left panel) and white (right panel), and addition of ABA did not increase 2-NBDG transport above control values. The observed increase of NBDG uptake in LANCL1/2 overexpressing brown and white adipocytes is in line with the increased expression of glucose transporter GLUT4 (Figure 4B,C, upper central panels). The metabolic fate of glucose in LANCL1/2-overexpressing brown and “beige” adipocytes is partly oxidative, as can be inferred from the increased transcription of the mitochondrial pyruvate transporter (MPC1) and of subunit alpha-1 of pyruvate dehydrogenase (PDHα1) (Figure 4B,C). An increase in triglycerides in LANCL1/2-overexpressing brown (Figure 9B, upper left panel) and white (Figure 9B, lower left panel) adipocytes also testifies to a biosynthetic exploitation of glucose in line with an increased expression of the insulin receptor. However, the increased transcription of the fatty acid mitochondrial transporter CPT1β in brown and “beige” adipocytes overexpressing LANCL1/2 also testifies to an increased mitochondrial oxidation of fatty acids. Transcriptional results thus suggest that both metabolic pathways, triglyceride synthesis and fatty acid oxidation, are upregulated in LANCL1/2-overexpressing white and brown adipocytes.

### 2.7. Chronic ABA Treatment Stimulates Transcription of Several Key Hormone Receptors and Enzymes Responsible for White Adipocyte “Browning” and Mitochondrial DNA Content in the Brown Adipose Tissue of LANCL2^−/−^ Mice

To investigate whether a “browning” effect of ABA on adipose tissue could occur in vivo, C57Bl/6 *LANCL2*^+/+^ (WT) and *LANCL2*^−/−^ (KO) mice (5/group) were fed a standard diet without (controls), or with ABA, administered in the drinking water, at a dose of approximately 1 μg/kg body weight (BW)/day. After 30 days, the animals were sacrificed and samples of interscapular BAT were taken to evaluate the expression of several genes.

Results obtained are shown in Figure 10. First of all, ablation of the LANCL2 protein in *LANCL2*^−/−^ mice was verified by evaluating mRNA levels (Figure 10, upper left panel). In LANCL2 KO mice, mRNA levels of LANCL1 were significantly higher relative to those in WT mice (Figure 10, upper right panel), as already observed in the skeletal muscle [9]. BAT mitochondrial DNA (MT-DNA) was lower in LANCL2 KO mice compared with WT; however, transcription of complex I of the respiratory chain (MT-ND1) was almost double compared with that in WT, as if to compensate for the reduced mitochondrial number. In ABA-treated WT mice, both the BAT mitochondrial DNA content (MT-DNA) and transcription of MT-ND1 increased significantly (1.7- and 4-fold, respectively), compared with that of untreated WT. In ABA-treated LANCL2 KO mice mitochondrial DNA was not significantly different compared with untreated KO, while MT-ND1 transcription increased significantly, although it did not reach levels similar to those of WT animals (Figure 10, central right panel). Transcription of the AMPK/PGC-1α/Sirt1 axis increased in the BAT from ABA-treated WT, but not from ABA-treated LANCL KO mice (Figure 10, central left panel), although mRNA levels in KO mice were basally significantly higher compared with WT, possibly a consequence of the increased LANCL1 expression levels in the BAT, as also observed in skeletal muscle [9].

Transcription of hormone receptors (ADRβ3, THRα1, THRβ) of DIO2, responsible for generation of the active hormone T3 from its precursor T4 directly inside target cells, and of ERRα, involved in the regulation of mitochondrial function and thermogenesis, was also explored in the BAT from ABA-treated WT and LANCL2 KO mice. All transcripts increased significantly in ABA-treated WT mice compared with controls, the highest increase being observed with ERRα (8-fold) (Figure 10, lower panel). Interestingly, BAT from LANCL2 KO mice showed significantly higher levels of all these transcripts compared with WT mice, although treatment with ABA resulted in a further slight increase of these levels. Nonetheless, transcription of ADRβ3, THRα1, THRβ, DIO2, and ERRα in the BAT from ABA-treated KO mice was only modestly lower than that in ABA-treated WT mice, likely due to their increased “basal” levels (Figure 10, lower panel).

## 3. Discussion

Collectively, results obtained in this study outline a hitherto unknown regulatory role for the ABA/LANCL1/2 hormone/receptor system in the physiology of human white and brown adipocytes. By activating the AMPK/PGC-1α/Sirt1 pathway both at the transcriptional and protein levels (Figure 1, Figure 2, Figure 3 and Figure 4A), LANCL proteins upregulate fundamental metabolic and mitochondrial functions of both brown and “beige” white adipocytes, pertaining to their energy-dissipating and thermogenic function.

As summarized in Figure 11, overexpression of LANCL1/2 stimulates while their combined silencing conversely significantly reduces glucose transport (via GLUT4 upregulation) and its oxidation (by upregulating transcription of MPC1 and PDHα1), mitochondrial biogenesis (MT-DNA), respiration (expression of complex I and basal and maximal O_2_ consumption) and proton gradient (∆Ψ) magnitude, expression of receptors for browning hormones (ADRβ3, THRα1/β), and of browning genes (UCP1/3). The higher mitochondrial respiration rate in LANCL1/2-overexpressing cells probably allows cells to maintain a higher ∆Ψ and ATP-linked respiration compared with control cells, both in white and brown adipocytes, despite the increased uncoupling.

LANCL1/2 overexpression per se stimulates all the above summarized metabolic and transcriptional functions; however, addition of ABA further increases this effect, allowing the conclusion that treatment with ABA exerts pro-browning effects on human adipocytes in vitro. Indeed, the uncoupling effect of ABA on white adipocytes overexpressing LANCL1/2 is more evident than that of T3, as demonstrated by the lower maximal and ATP-linked respiration in ABA-treated vs. untreated adipocytes (Figure 6B, upper panels).

A similar pro-browning effect of ABA is also evident on the brown adipose tissue of mice treated for 4 weeks with ABA, which shows a 2- and 4-fold increase of mitochondrial DNA and OXPHOS complex I (MT-ND1), respectively (Figure 10). In addition, the upregulation of “browning” hormones’ receptors (ADRβ3, THRα1, THRβ, and ERRα) observed on human adipocytes also occurs on adipocytes from ABA-treated mice (Figure 10). In the BAT from LANCL2 KO mice, which show significantly lower levels of MT-DNA than wild-type animals, expression of MT-ND1 is instead higher than in WT and ABA treatment, which further increases MT-ND1 to levels similar to those observed in ABA-treated WT (Figure 10). Interestingly, expression of thyroid hormone receptors and of ERRα is also constitutively higher in LANCL2 KO mice compared to their WT siblings. An increased sensitivity to thyroid hormones and an enhanced ERRα activity in the BAT from LANCL2 KO mice may be the reason for the increased transcription of MT-ND1, which may partly compensate for the reduced MT-DNA by making mitochondria more performing. These results on LANCL2 KO mice suggest a specific role for LANCL1 in upregulating transcription of MT-ND1, THR, and ERRα. Indeed, expression of LANCL1 in adipose tissue is approx. 2-fold higher compared with LANCL2 (Supplementary data).

Indeed, LANCL1 is strikingly upregulated in the BAT from LANCL2 KO as compared with WT mice (Figure 10) to a higher degree as previously observed in the skeletal muscle, which showed a 4-fold increase of LANCL1 expression [9]. These observations point to a transcriptional link between LANCL1 and LANCL2, allowing the increase of one to compensate for the reduction of the other. In addition, a reciprocal positive transcriptional effect of the LANCL proteins is evident in human adipocytes overexpressing LANCL1/2, where mRNA levels of both proteins keep increasing during adipocyte differentiation (up to 30-fold control levels, in PLV-infected cells, for differentiated white adipocytes, Figure 4C central lower panel), despite the fact that overexpression of both proteins occurs via retroviral infection. Apparently, increased protein levels of retrovirally-encoded LANCL1/2 increase transcription of the endogenous LANCL genes too.

Another transcriptional link highlighted by this study is the one between the LANCL1/2 proteins and receptors and enzymes responsible for the adipocyte response to major hormonal stimuli regulating adipocyte energy metabolism and thermogenesis, i.e., the beta adrenergic receptor (ADRβ3), the thyroid hormone receptors (THR) and the enzyme deiodinase (DIO2), which allows conversion of T4 to the ten-fold more active T3 inside target cells. Upregulation, or conversely downregulation of mRNAs for these hormonal receptors in LANCL1/2-overexpressing or -silenced cells clearly establishes a hierarchical role of the LANCL proteins over these key regulators of adipocyte function.

Finally, a hitherto unknown transcriptional regulation by the LANCL1/2 proteins of ERRα is demonstrated in human white and brown adipocytes: overexpression of LANCL1/2 increases 20-fold, while their combined silencing reduces ERRα mRNA levels in both white and brown adipocytes (Figure 4B,C and Figure 5B,C, respectively). In addition, ABA simulates ERRα expression in the BAT of WT mice and ERRα is spontaneously overexpressed in the BAT of LANCL2 KO mice, which overexpress LANCL1 (Figure 10). ERRα is an orphan receptor which plays a pivotal role in adipocyte energy metabolism and mitochondrial function. ERRα is part of the AMPK/PCG1α axis that controls mitochondrial biogenesis, respiration, and thermogenesis [26] and that is also transcribed at higher levels in the BAT from LANCL2 KO vs. WT mice (Figure 10). ERRα, together with its homolog ERRγ, mediates the complex transcriptional response of BAT to adrenergic stimulation, in turn stimulated by cold. As a result, mice lacking ERRα/γ have a severely impaired oxidative and thermogenic capacity and rapidly become hypothermic when exposed to cold [27]. Thus, the fact that ERRα levels are in turn controlled by the ABA/LANCL system adds a new and significant link to a signaling chain of pivotal physiological importance in the mammalian response to cold stress. Interestingly, ERRα lies downstream of both PGC-1α and HDAC3, a histone deacetylase which stimulates transcription of PGC-1α, UCP1, and OXPHOS genes in BAT [28] and which has recently been shown to regulate a futile cycle of fatty acid synthesis/oxidation key to white adipocytes browning [29]. It is tempting to speculate that a metabolic futile cycle of triglyceride synthesis (from glucose) and oxidation may fuel in part the increased oxidation rate of LANCL1/2-overexpressing beige and brown human adipocytes.

In conclusion, this study identifies the ABA/LANCL1-2 hormone/receptors system as a new player in the control of human brown and “beige” mitochondrial respiration and uncoupling and provides the basis for future in vitro and in vivo studies. The fact that LANCL1/2 overexpression per se increases transcription of β-adrenergic and thyroid hormone receptors in brown and white human differentiated adipocytes is expected to increase cell sensitivity to their uncoupling effect. In addition, dietary intake of ABA, e.g., provided by vegetal extracts, is expected to stimulate white adipocyte “browning” and increase adipocyte energy expenditure, an action that should result in reduced body weight gain under conditions of high-calories dietary regimens. Finally, the interplay between the ABA/LANCL system and other hormones involved in body energy balance (T3 and catecholamines) is open to investigation.

## 4. Materials and Methods

### 4.1. Cell Culture

TERT-immortalized polyclonal brown (TERT-hBA) and white (TERT-hWA) preadipocytes were kindly donated by Dr. Jacob B. Hansen from the Department of Biology of the University of Copenhagen (Copenhagen, Denmark) [17] and cultured in Advanced DMEM/F12 supplemented with 10% FBS, L-glutamine (2 mM), penicillin (62.5 μg/mL), streptomycin (100 μg/mL), and basic fibroblast growth factor (βFGF) (2.5 ng/mL). The cells were kept at 37 °C in a humidified atmosphere with 5% CO_2_.

The choice of an ABA concentration of 100 nM for all in vitro experiments derives from the following considerations: (i) this concentration has been shown to be effective on skeletal myocytes and on cardiomyocytes, stimulating oxidative metabolism, mitochondrial respiration and uncoupling [9,16]; (ii) normal plasma ABA is in the low nanomolar range in humans [5], and it increases 20–60 times after dietary intake of ABA-rich fruits or vegetal extracts [6]. Thus, a concentration of 100 nM for ABA was chosen in the present study because it is both effective in vitro on previously explored cell types and also attainable in vivo.

### 4.2. Lentiviral and Retroviral Cell Transduction

The lentiviral plasmids pLV[shRNA]-Puro-U6 encoding for a control scramble shRNA (shRNA-SCR), for the shRNA targeting human LANCL1 (shRNA-L1) and for the shRNA targeting human LANCL2 (shRNA-L2), were purchased from Vector Builder (Chicago, IL, USA). Overexpression of human LANCL1 (ovLANCL1) and of human LANCL2 (ovLANCL2) was obtained in TERT-hBA and TERT-hWA pre-adipocytes using pBABE vectors constructed as described in [9], with the empty vector pBABE (Addgene, MA, USA) as negative control (PLV). To obtain pre-adipocytes stably silenced for, or overexpressing LANCL1 and LANCL2, the following protocols were used. For silencing of LANCL1 and LANCL2, Lentiviral Vector Particles (LVPs) were generated in HEK-293T cells. Briefly, HEK-293T cells were seeded (3 × 10^5^ cells on 6-cm plates) in Dulbecco’s modified Eagle medium, 10% FBS and 1% penicillin-streptomycin. After 24 h, cells were cotransfected with a Δ891 and a VSV-G encoding vector along with a shRNA transfer lentiviral plasmid (in parallel, shRNA-SCR, shRNA-L1 and shRNA-L2) using TransIT^®^ Transfection Reagent (Mirus, Madison, WI, USA). After 24 h, the HEK-293T medium was changed with Dulbecco’s modified Eagle medium, 10% FBS and 1% penicillin-streptomycin to promote viral production. The supernatant containing lentiviral particles was collected 48 and 72 h after transfection, filtered with a 0.45-μm-diameter filter and used to infect pre-adipocytes (1 × 10^6^ cells on 10-cm plates) in the presence of protamine sulfate (final concentration 5 μg/mL). After the second cycle of infection, cells were selected with puromycin (2 μg/mL). The knockdown efficiency was validated by evaluating LANCL1 and LANCL2 mRNA and protein levels by qPCR and Western blot analysis, respectively. For overexpression of LANCL1 and LANCL2, we used the same method described above, except for the plasmids (pBABE vectors) and the cells used for retroviral packaging (HEK-Plat-A cells). Overexpression efficiency was assessed using Western blot analysis.

### 4.3. qPCR Analysis

qPCR analyses were performed on cDNA samples from immortalized human adipocytes and from the BAT of LANCL2 KO mice. After sacrifice, samples of interscapular BAT (approximately 30 mg) were taken from mice chronically treated with or without ABA for 4 weeks (1 μg/BW/day administered in the drinking water). Brown and white pre-adipocytes were instead incubated with or without 100 nM ABA for 4 h. Total RNA was extracted from brown adipose tissue of mice using QIAzol Lysis Reagent and Tissue Lyser instrument (Qiagen, Milan, Italy) and from human pre-adipocytes using RNeasy Micro Kit (Qiagen, Milan, Italy) according to the manufacturer’s instructions. The cDNA was synthesized by using iScript cDNA Synthesis Kit (Bio-Rad, Milan, Italy) starting from 1 μg of total RNA and was used as a template for qPCR analysis: reactions were performed in an iQ5 Real-Time PCR detection system (Bio-Rad, Milan, Italy). The mouse- and human-specific primers were designed using Beacon Designer 2.0 software (Bio-Rad, Milan, Italy) and their sequences are listed in Appendix A (primer sequences used to amplify mouse target genes) and Appendix A (primer sequences used to amplify human target genes). Each sample was assayed in triplicate in a 25 μL amplification reaction, containing 4 ng of cDNA, primers mixture (0.4 μM each of sense and antisense primers) and 12.5 μL of 2X iQ SYBR Green Supermix Sample (Bio-Rad, Milan, Italy). The amplification program included 40 cycles of two steps, each comprising heating to 95 °C and to 62 °C, respectively. Fluorescence products were detected at the last step of each cycle. To verify the purity of the products, a melting curve was produced after each run. Values for human and mouse genes were normalized on hypoxanthine-guanine phosphoribosyltransferase-1 (Hprt1) mRNA expression. Statistical analysis of the qPCR was performed using the iQ5 Optical System Software version 1.0 (Bio-Rad Laboratories) based on the 2^−∆∆Ct^ method [9]. The dissociation curve for each amplification was analyzed to confirm the absence of nonspecific PCR products.

### 4.4. Western Blot

Brown and white human preadipocytes (1 × 10^6^/well) were seeded in 6-well plates. After 24 h, the supernatant was removed, cells were washed once in Krebs-Ringer HEPES buffer (KRH) and then incubated in KRH with 5 mM glucose for 60 min at 37 °C with or without 100 nM ABA. After removal of the supernatant, the cells were scraped in a volume of 300 μL lysis buffer (20 mM Tris-HCl pH 7.4, 150 mM NaCl, 1 mM EDTA, 1% NP40) containing a protease inhibitor cocktail. After brief sonication, the protein concentration was determined on an aliquot of each lysate. Lysates (30 μg) were loaded on 10% polyacrylamide gel and separated with SDS-PAGE and proteins were transferred to nitrocellulose membranes (Bio-Rad, Milan, Italy) according to standard procedures. The membranes were blocked for 1 h with 20 mM Tris-HCl pH 7.4, 150 mM NaCl, and 1% Tween 20 (TBST) containing 5% non-fat dry milk and incubated for 1 h at room temperature with primary antibodies (Appendix A). Following incubation with the appropriate secondary antibodies (Appendix A) and ECL detection (GE Healthcare, Milan, Italy), band intensity was quantified with the ChemiDoc imaging system (Bio-Rad, Milan, Italy).

### 4.5. Adipocyte Differentiation

TERT-hBA and TERT-hWA cells were seeded at 1 × 10^6^ per well in 6-well plates. At two-day post confluence (designated day 0), adipogenesis was induced with a differentiation cocktail containing Advanced DMEM/F12 containing 2% FBS, 2 mM L-glutamine, 62.5 μg/mL penicillin, 100 μg/mL streptomycin (basal medium), and supplemented with insulin (5 μg/mL), dexamethasone (1 μM), 3-isobutyl-1-methylxanthine (IBMX) (0.5 mM), rosiglitazone (1 μM), human cortisol (1 μM), and T3 (1 nM). On day 3, the medium was refreshed with the same medium used at day 0. Between days 6 and 12, TERT-hWA were cultured in basal medium to reach full white differentiation (day 12). For “browning” of white adipocytes, TERT-hWA were further cultured for 3 days, until day 15, in the presence of 1 μM rosiglitazone. Conversely, between days 6 and 12 TERT-hBA were differentiated to brown adipocytes by addition of 1 nM T3 to the basal culture medium [17]. Appendix A in the supplementary data summarizes the differentiation protocol. At day 6, approximately 60% of cells already showed an adipocyte morphology, with accumulation of lipid droplets, while at day 10 this percentage increased to about 80–90%. During white or brown differentiation, TERT-hBA and TERT-hWA cells were treated or not (controls) with 100 nM ABA from day 0 to day 12; during the “browning” process of white adipocytes (days 12–15), differentiated TERT-hWA cells were treated with or without 100 nM ABA, 1 μM rosiglitazone or both.

### 4.6. Oil Red O Staining

To measure cellular neutral lipid droplet accumulation, differentiated TERT-hBA and TERT-hWA adipocytes were washed three times with iced Phosphate-Buffered Saline (PBS) and fixed with 4% paraformaldehyde for 30 min. After fixation, cells were washed three times and stained with Oil Red O solution (0.5 g Oil Red O dissolved in 60% ethanol) for 15 min at room temperature. Cells were washed again three times with PBS to remove excess staining. Adipocytes were examined under an inverted phase contrast light microscope: two different microscopic fields (20× and 40× magnifications) per culture were photographed. To obtain a more quantitative measure of triglyceride accumulation, the red oil droplets in the cells were extracted in 1 mL of 100% isopropanol and the absorbance of the solution was measured at 510 nm with a spectrophotometer.

### 4.7. Glucose Transport Assays

Differentiated adipocytes overexpressing or silenced for the expression of both LANCL1 and LANCL2 were cultured overnight at 5 × 10^3^/well in a 96-well plate in 5 mM DMEM without serum. Cells were washed once with DMEM and then incubated for 5 min at 37 °C in DMEM containing 100 nM ABA. At the end of incubation, cells were washed with KRH at 37 °C. The fluorescently labeled deoxyglucose analog 2-NBDG (50 μM) was added to each well and, after 15 min, the supernatant was removed, wells were washed once with ice-cold KRH, 50 μL KRH was added to each well and the mean fluorescence (l_ex_ = 465 nm, l_em_ = 540 nm) from 10 acquisitions/well was calculated. Each experimental condition was assayed in at least 8 wells. Unspecific 2-NBDG uptake, determined in the presence of the glucose transport inhibitors cytochalasin B (20 mM) and phloretin (200 mM) [10], was subtracted from each experimental value.

### 4.8. JC-1 Analysis

Differentiated TERT-hBA and TERT-hWA adipocytes overexpressing human LANCL1 and human LANCL2 or silenced for the expression of both human LANCL1 and human LANCL2 with the respective negative controls (PLV and shRNA-SCR) were stained with the cationic dye JC-1 (Thermo Fisher Scientific, Waltham, MA, USA), which exhibits potential-dependent accumulation in mitochondria. At low membrane potentials, JC-1 exists as a monomer and produces a green fluorescence (l_em_ = 527 nm), whereas at high membrane potentials JC-1 forms aggregates and produces a red fluorescence (l_em_ = 590 nm). Thus, mitochondrial depolarization is indicated by a decrease in the red/green fluorescence intensity ratio [25]. Briefly, fully differentiated adipocytes were seeded at 3 × 10^4^ onto µ-slide wells, treated or not with 100 nM ABA for 4 h, stained with JC-1 (2.5 µg/mL) for 20 min at 37 °C in a 5% CO_2_ incubator and then imaged live. The red/green ratio was analyzed after a background subtraction with the ImageJ software (v1.8.0, National Institutes of Health, Bethesda, MD, USA), using a quantitative analysis based on an intensity measurement of specific selected ROIs.

### 4.9. Immunofluorescence

Differentiated adipocytes overexpressing or silenced for the expression of both LANCL proteins were plated on glass coverslips, treated for 4 h with 100 nM ABA and then stained with MitoTracker Deep Red FM (Invitrogen ^TM^, Life Technologies, Carlsbad, CA, USA). Briefly, 50 μg of lyophilized MitoTracker was dissolved in 100 μL of DMSO to produce 0.92 mM stock (kept in −20 °C). A fresh staining solution (100 nM) was prepared each time before use by diluting the stock solution in serum-free medium. The cells were stained for 40 min at 37 °C in a 5% CO_2_ incubator and then imaged live. Images were acquired on a Leica TCS SP confocal laser scanning microscope, equipped with 476, 488, 543 and 633 excitation lines with a 60× Plan Apo oil objective. The MitoTraker Red signal was then analyzed, after background subtraction, using ImageJ 1,46 software (Wayne Rasband, National Institutes of Health, Bethesda, MD, USA) using a semi-manual method based on the delimitation of the cells.

### 4.10. Seahorse Analysis

Oxygen consumption rate (OCR) was determined using a Seahorse XFp Extracellular Flux Analyzer (Agilent Technologies, Santa Clara, CA, USA). On day 9 of differentiation, TERT-hBA and TERT-hWA adipocytes were replated in XF plates at a density of 5000 cells per well. At day 12 or 15, differentiated cells were incubated at 37 °C for 45 min in no-CO_2_ incubator with Agilent Seahorse DMEM pH 7.4, enriched with glucose (17.5 mM), glutamine (2 mM) and pyruvate (1 mM). The bioenergetic profile was measured using the Cell Mito Stress Test Kit (Cat. #103010-100) according to the manufacturer’s instructions. Three measurements of OCR were taken under control conditions and after sequential injections of interest compound (ABA (100 nM), T3 (100 nM)), oligomycin (1.5 µM, ATP synthase inhibitor), carbonyl cyanide-4-(trifluoromethoxy) phenylhydrazone (FCCP, 1.5 µM, proton gradient dissipation) and rotenone (0.5 µM, respiratory Complex I inhibitor) plus antimycin A (0.5 µM, respiratory Complex III inhibitor). OCR was normalized to total cellular protein content determined directly in the plate, immediately after each experimental run, by Bradford assay [30] and reported as pmol/min/µg protein.

### 4.11. In Vivo Experiments

LANCL2^−/−^ C57Bl/6 mice and their wild-type siblings, obtained from heterozygous breeding [10], were housed at the animal facility of the IRCCS San Martino Hospital (Genova, Italy). All mice used in this study were derived from a heterozygous LANCL2^+/−^ x LANCL2^+/−^ breeding scheme. Genotyping of the offspring allowed the selection of LANCL2^+/+^ (henceforth referred to as control or wild-type or WT) and LANCL2^−/−^ (KO) mice. Protocols of animal use were approved by the Italian Ministry of Health, in line with the EU Directive 2010/63/EU for animal experiments. Male, six week-old C57Bl/6 *LANCL2*^+/+^ (WT) and *LANCL2*^−/−^ (KO) mice (5/group) were fed a standard diet and were treated, or not (controls), with ABA, administered through the drinking water. The concentration of ABA in the water was calculated taking into consideration the average daily volume introduced by each mouse (≅4 mL), so as to reach a daily dose of approximately 1 μg/kg BW of ABA. After 30 days, the animals were sacrificed and samples of 30 mg of interscapular brown adipose tissue were excised and immediately processed for RNA extraction for qPCR analysis.

### 4.12. DNA Extraction and Determination of Mitochondrial Levels by qPCR Analysis

Total DNA was extracted from human brown and white adipocytes (1 × 10^6^ cells on 6-cm plates) and from brown adipose tissue of mice (approximately 10 mg) with the QIAamp DNA Micro Kit (Qiagen, Milan, Italy) according to the manufacturer’s protocol. The purity and quantity of DNA were evaluated with the NanoDrop 1000 (Thermo Fisher Scientific, Waltham, MA, USA). The mitochondrial/genomic DNA ratio was determined using specific primers listed in Appendix A for mitochondrial ND1 (MT-ND1) or tRNA^Leu^ (MT-DNA) and genomic/nuclear HPRT1 by qPCR analysis as described in [Section 4.3].

### 4.13. Statistical Analysis

The results were expressed as mean ± SD. Statistical analysis was performed using the GraphPad Prism 7 Software (GraphPad Software, San Diego, California, CA, USA). Comparisons were drawn by unpaired, two-tailed Student’s *t*-test. A value of *p* < 0.05 was considered to be statistically significant.

## Figures and Tables

**Figure 1 ijms-24-03489-f001:**
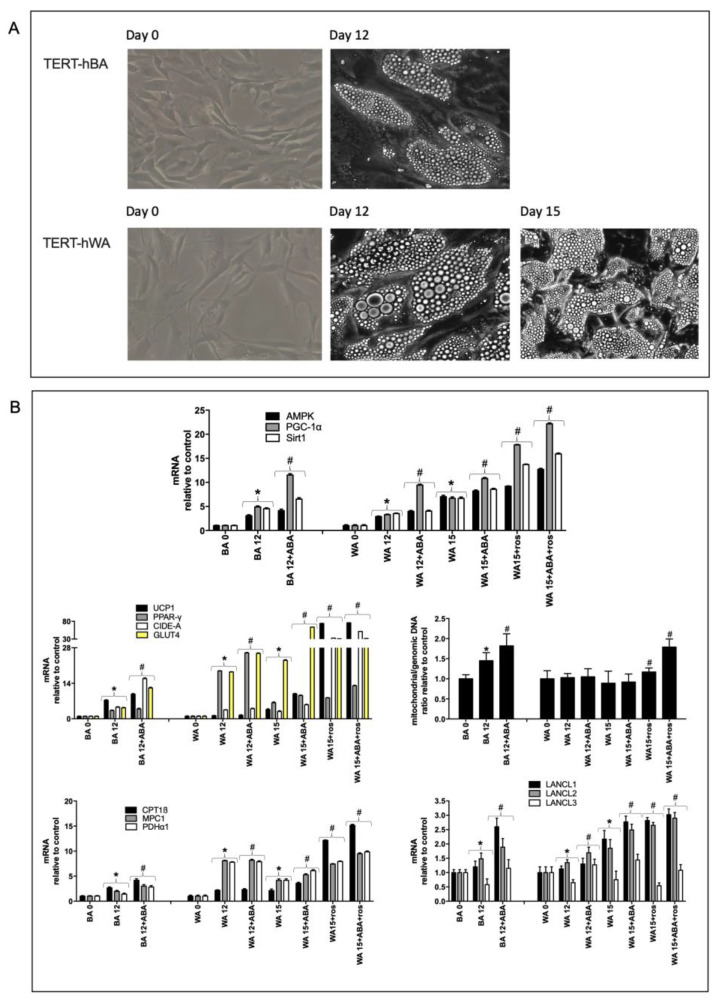
Effect of ABA on human white and brown adipocyte differentiation. (**A**) representative micrographs (magnification ×20×40) of immortalized brown (TERT-hBA, upper panels) and white (TERT-hWA, lower panels) human preadipocytes at days 0, 12 and 15 of differentiation. (**B**) qPCR analysis; human TERT-hWA and TERT-hBA preadipocytes differentiated to adipocytes in the absence (control) or presence of 100 nM ABA and/or 1 μM rosiglitazone: at the indicated time points (day 0, 12 or 15), the mRNA levels of the indicated proteins were evaluated by qPCR. Upper panel, AMPK-1α, PGC-1α and Sirt1 mRNAs; central left panel, UCP1, PPAR-γ, CIDE-A and GLUT4 mRNAs; central right panel, mitochondrial/genomic DNA ratio (MT-DNA); lower left panel, CPT1β, MPC1 and PDHα1 mRNAs; lower right panel, LANCL1, LANCL2 and LANCL3 mRNAs. Results shown are the mean ± SD from at least 4 experiments; * *p* < 0.04 relative to undifferentiated untreated control, # *p* < 0.004 relative to differentiated control. *p* values are calculated by unpaired, two-tailed *t*-test.

**Figure 2 ijms-24-03489-f002:**
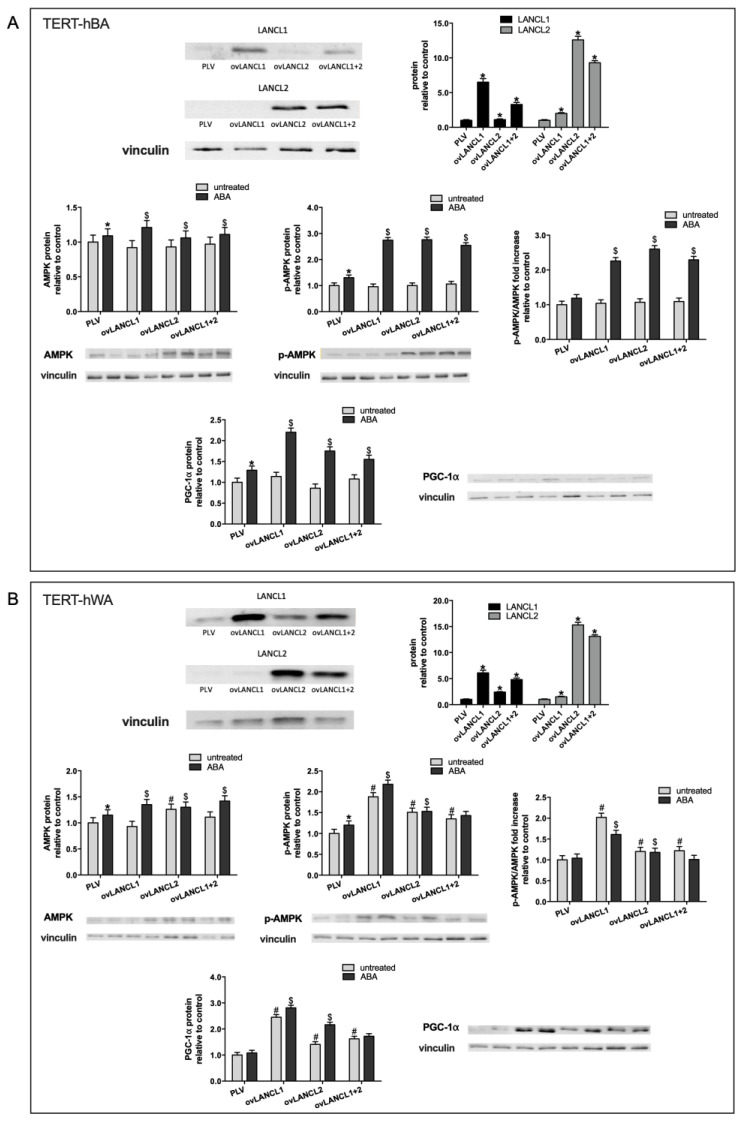
Overexpression of LANCL1/2 proteins increases total and phosphorylated AMPK and PGC-1α protein levels in brown (TERT-hBA) and white (TERT-hWA) human preadipocytes. LANCL1 and/or LANCL2 were stably overexpressed in human TERT-hBA (**A**) or TERT-hWA (**B**) pre-adipocytes by viral infection. Upper left panels, representative Western blots of LANCL1/2 proteins in cells overexpressing LANCL1 (ovLANCL1), LANCL2 (ovLANCL2) or both LANCL1 and LANCL2 (ovLANCL1+2); upper right panels, densitometric quantitation of the LANCL proteins relative to control cells, transfected with the empty vector (PLV); total AMPK, phosphorylated (Ser473) AMPK (pAMPK), p-AMPK/AMPK ratio and PGC-1α in cells overexpressing LANCL1 (ovLANCL1), LANCL2 (ovLANCL2) or both LANCL1 and LANCL2 (ovLANCL1+2), relative to PLV. The order of the samples is the same in the quantifications and in the representative blots except for the lower right and central panels of Figure 2A which show the first 4 bands for untreated samples and the last 4 bands for ABA-treated samples. Values are normalized against vinculin, as housekeeping protein; * *p* < 0.03 and # *p* < 0.004 relative to untreated control; $ *p* < 0.005 relative to ABA-treated PLV-infected cells. *p* values are calculated by unpaired, two-tailed *t*-test.

**Figure 3 ijms-24-03489-f003:**
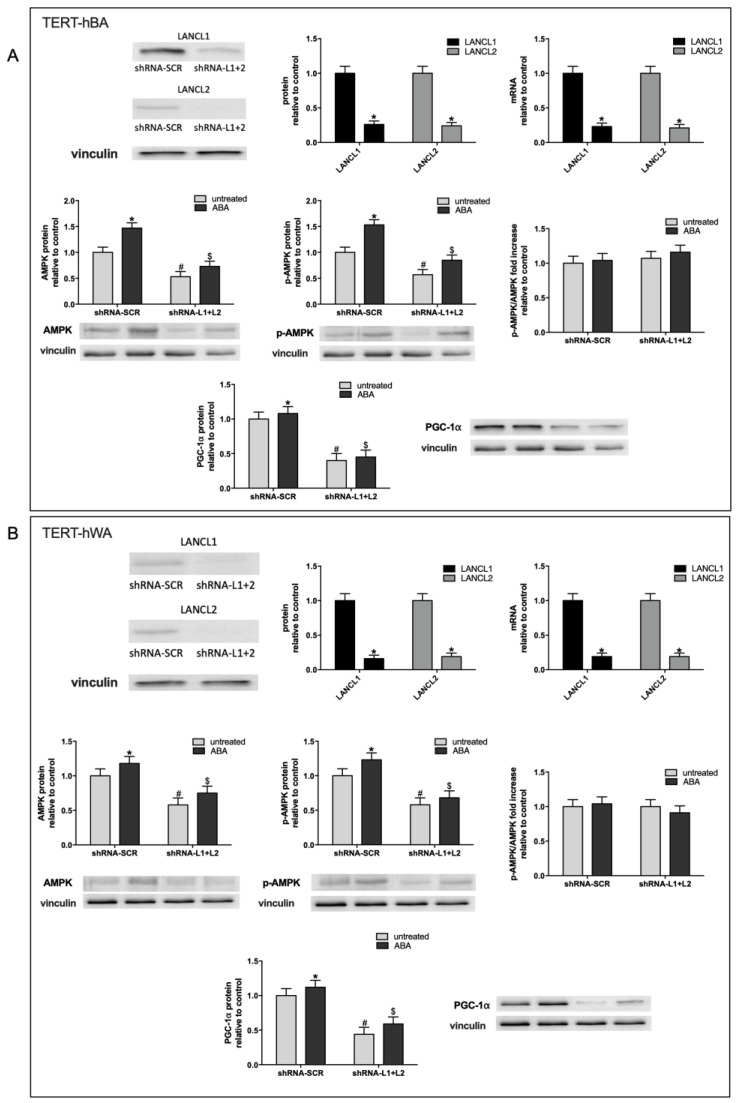
The combined silencing of LANCL1/2 proteins reduces total and phosphorylated AMPK and PGC-1α protein levels in brown (TERT-hBA) and in white (TERT-hWA) preadipocytes. LANCL1 and LANCL2 were stably silenced in human immortalized TERT-hBA (**A**) or TERT-hWA (**B**) pre-adipocytes by viral infection. Upper left panels, representative Western blots of LANCL1/2 proteins in cells silenced for both LANCL1 and LANCL2 (shRNA-L1+L2); upper central panels, densitometric quantitation of the LANCL proteins relative to control cells, transfected with the vector containing scrambled silencing sequences (shRNA-SCR); upper right panels, LANCL1/2 mRNA levels in LANCL1/2-silenced cells relative to control; total AMPK, phosphorylated (Ser473) AMPK (pAMPK), p-AMPK/AMPK ratio and PGC-1α in double-silenced cells, relative to shRNA-SCR. Values are normalized against vinculin, as housekeeping protein; * *p* < 0.01 and # *p* < 0.002 relative to untreated control; $ *p* < 0.005 relative to ABA-treated shRNA-SCR-infected cells. *p* values are calculated by unpaired, two-tailed *t*-test.

**Figure 4 ijms-24-03489-f004:**
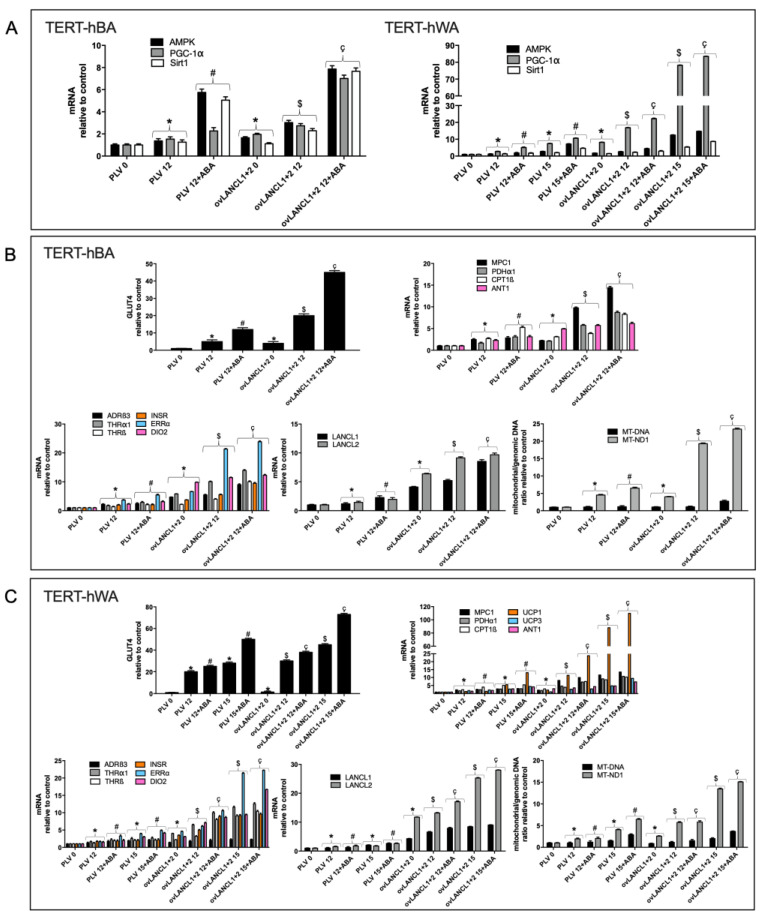
Transcriptional effects of the overexpression of LANCL1/2 on differentiated brown and white adipocytes. Human brown (TERT-hBA) or white (TERT-hWA) preadipocytes overexpressing LANCL1 and LANCL2 (ovLANCL1+2), or control cells infected with the empty vector (PLV), were differentiated to white or brown adipocytes in the absence or in the presence of 100 nM ABA: at days 0, 12 and 15 of culture, mRNA levels of the indicated genes were evaluated by qPCR. (**A**) analysis of the AMPK/PGC-1α/Sirt1 signaling axis in TERT-hBA (left panel) and TERT-hWA (right panel). (**B**) TERT-hBA; upper left panel, GLUT4 mRNA; upper right panel, carnitine palmitoyl transferase 1β (CPT1β), mitochondrial pyruvate carrier 1 (MPC1), pyruvate dehydrogenase subunit α1 (PDHα1) and adenine nucleotide translocase 1 (ANT1) mRNAs; lower left panel, adrenergic receptor β-3 (ADRβ3), thyroid receptor α-1 (THRα1), thyroid receptor β (THRβ), insulin receptor (INSR), estrogen related receptor α (ERRα) and deiodinase 2 (DIO2) mRNAs; lower central panel, LANCL1/2 mRNAs; lower right panel, mitochondrial/genomic DNA ratio (MT-DNA and MT-ND1). * *p* < 0.02 relative to undifferentiated untreated control, # *p* < 0.006 relative to differentiated control, $ *p* < 0.003 relative to undifferentiated untreated overexpressing LANCL1 and LANCL2 control, ç *p* < 0.001 relative to differentiated overexpressing LANCL1 and LANCL2 control. (**C**) TERT-hWA; upper left panel, GLUT4 mRNA; upper right panel, CPT1β, MPC1, PDHα1, UCP1, UCP3 and ANT1 mRNAs; lower left panel, ADR β3, THRα1, THRβ, INSR, ERRα and DIO2 mRNAs; lower central panel, LANCL1/2 mRNAs; lower right panel, mitochondrial/genomic DNA ratio (MT-DNA and MT-ND1). Results shown are the mean ± SD from at least 5 experiments; * *p* < 0.02 relative to undifferentiated, untreated control cells, # *p* < 0.006 relative to differentiated control cells, $ *p* < 0.003 relative to undifferentiated, untreated cells overexpressing LANCL1/2, c, *p* < 0.001 relative to differentiated cells overexpressing LANCL1/2. *p* values are calculated by unpaired, two-tailed *t*-test.

**Figure 5 ijms-24-03489-f005:**
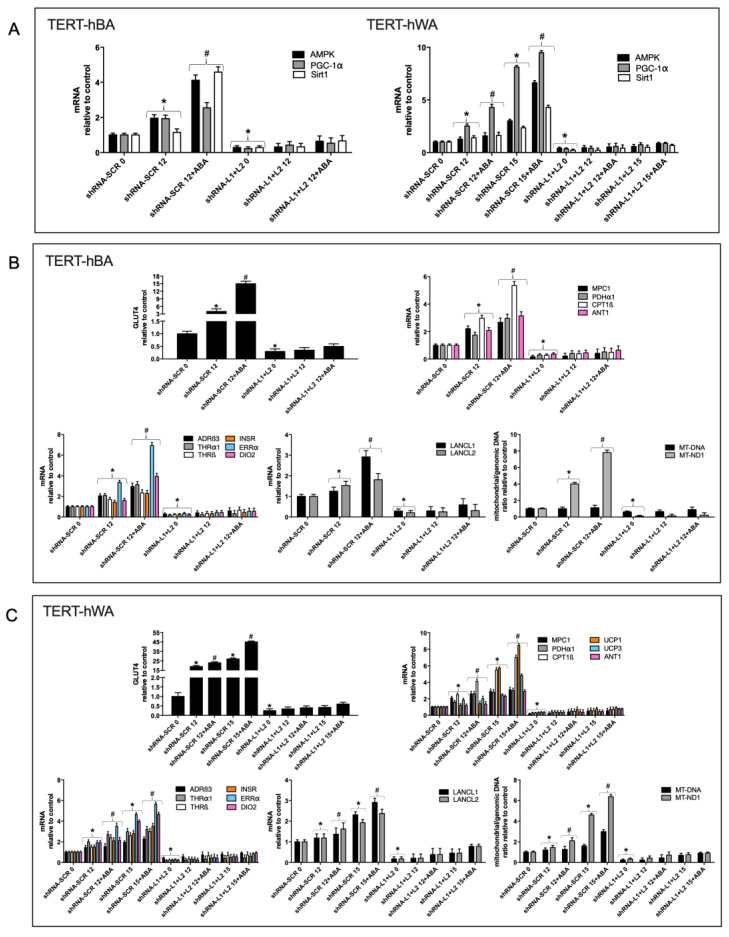
Transcriptional effects of the combined silencing of LANCL1/2 on differentiated brown and white adipocytes. Human brown (TERT-hBA) or white (TERT-hWA) preadipocytes silenced for the expression of LANCL1 and LANCL2 (shRNA-L1+L2), or control cells infected with the vector containing scrambled sequences (shRNA-SCR), were differentiated to white or brown adipocytes in the absence or in the presence of 100 nM ABA: at days 0, 12 and 15 of culture, mRNA levels of the indicated genes were evaluated by qPCR. (**A**) analysis of the AMPK/PGC-1α/Sirt1 signaling axis in TERT-hBA (left panel) and TERT-hWA (right panel). (**B**) TERT-hBA; upper left panel, GLUT4 mRNA; upper right panel, CPT1β, MPC1, PDHα1 and ANT1 mRNAs; lower left panel, ADRβ3, THRα1, THRβ, INSR, ERRα and DIO2 mRNAs; lower central panel, LANCL1/2 mRNAs; lower right panel, mitochondrial/genomic DNA ratio (MT-DNA and MT-ND1). * *p* < 0.02 relative to undifferentiated untreated control, # *p* < 0.006 relative to differentiated control. (**C**) TERT-hWA; upper left panel, GLUT4 mRNA; upper right panel, CPT1β, MPC1, PDHα1, UCP1, UCP3 and ANT1 mRNAs; lower left panel, ADRβ3, THRα1, THRβ, INSR, ERRα and DIO2 mRNAs; lower central panel, LANCL1/2 mRNAs; lower right panel, mitochondrial/genomic DNA ratio (MT-DNA and MT-ND1). Results shown are the mean ± SD from at least 5 experiments; * *p* < 0.02 relative to undifferentiated, untreated control cells, # *p* < 0.006 relative to differentiated control cells. *p* values are calculated by unpaired, two-tailed *t*-test.

**Figure 6 ijms-24-03489-f006:**
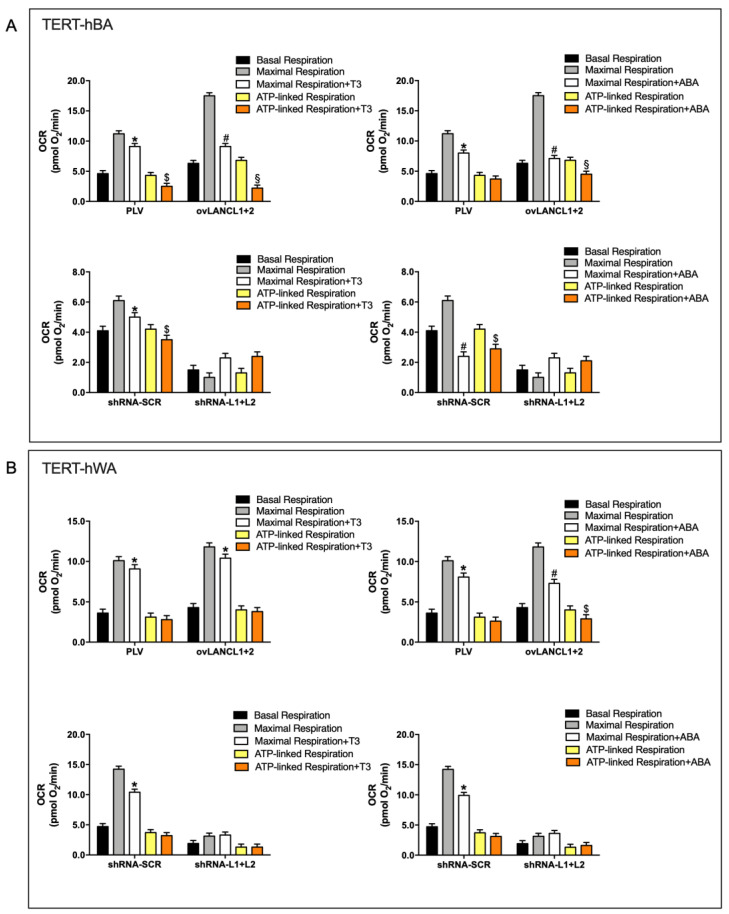
Mitochondrial respiration and uncoupling are controlled by LANCL1/2 expression levels in differentiated white and brown adipocytes. White (TERT-hWA) and brown (TERT-hBA) adipocytes, either overexpressing LANCL1/2 (ovLANCL1+2) and their controls, infected with the empty vector (PLV), or silenced for LANCL1/2 (shRNA-L1+L2) and their controls, infected with the scrambled silencing sequences (shRNA-SCR) were differentiated (up to day 12 of culture for brown adipocytes and up to day 15 of culture for white-derived “beige” adipocytes). Respiration measurements were performed using the Seahorse XFp Analyzer, with the sequential addition of oligomycin, FCCP and rotenone/antimycin A. Maximal and ATP-linked oxygen consumption rates (OCR) were measured in the absence or presence of 100 nM T3 or 100 nM ABA. (**A**) OCR of TERT-hBA; upper panels, LANCL1/2-overexpressing adipocytes, treated with T3 (**left**) or with ABA (**right**); lower panels, LANCL1/2-silenced adipocytes, treated with T3 (**left**) or with ABA (**right**). (**B**) OCR of TERT-hWA; upper panels, LANCL1/2-overexpressing adipocytes, treated with T3 (**left**) or with ABA (**right**); lower panels, LANCL1/2-silenced adipocytes, treated with T3 (**left**) or with ABA (**right**). * *p* < 0.05 and # *p* < 0.01 relative to maximal respiration without treatment (gray bar); $ *p* < 0.05 and § *p* < 0.01 relative to ATP-linked respiration without treatment (yellow bar), by unpaired *t*-test. Data shown are the mean ± SD of 3 experiments per group, with each value calculated in triplicate.

**Figure 7 ijms-24-03489-f007:**
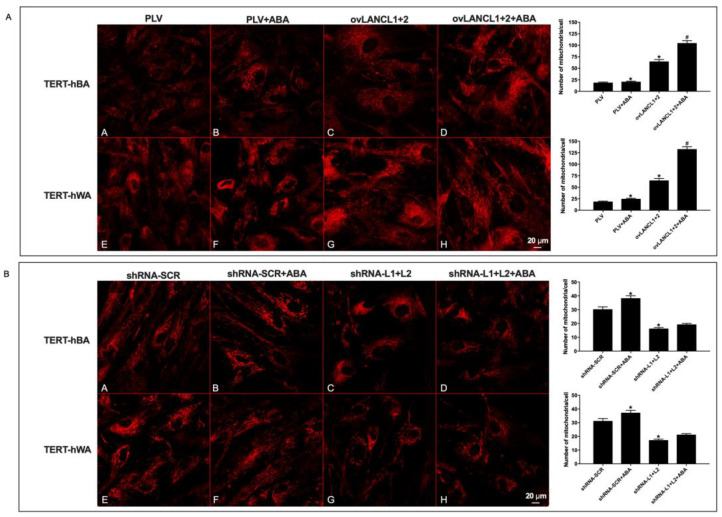
Mitochondrial number increases in LANCL1/2-overexpressing, and is conversely reduced in LANCL1/2-silenced white and brown adipocytes. The mitochondrial number was evaluated by MitoTracker analysis in TERT-hBA and TERT-hWA cells overexpressing LANCL1 and LANCL2 (ovLANC1+2) or double-silenced for the expression of both proteins (shRNA-L1+L2), and their respective controls, cells infected with the empty vector (PLV) or with the scrambled silencing sequences (shRNA-SCR), treated or not with 100 nM ABA. Representative confocal microscopy images of brown (panels A–D) and white (panels E–H)) adipocytes overexpressing (**A**) or silenced (**B**) for LANCL1 and LANCL2. The bar diagrams show the mean number of mitochondria per cell in each experiment (approx. 10 cells analyzed/cell type). The mean ± SD of the relative mitochondrial fluorescence was always calculated in at least 3 microscopic fields. * *p* < 0.005 relative to untreated control cells and # *p* < 0.01 relative to ABA-treated control cells by unpaired *t*-test.

**Figure 8 ijms-24-03489-f008:**
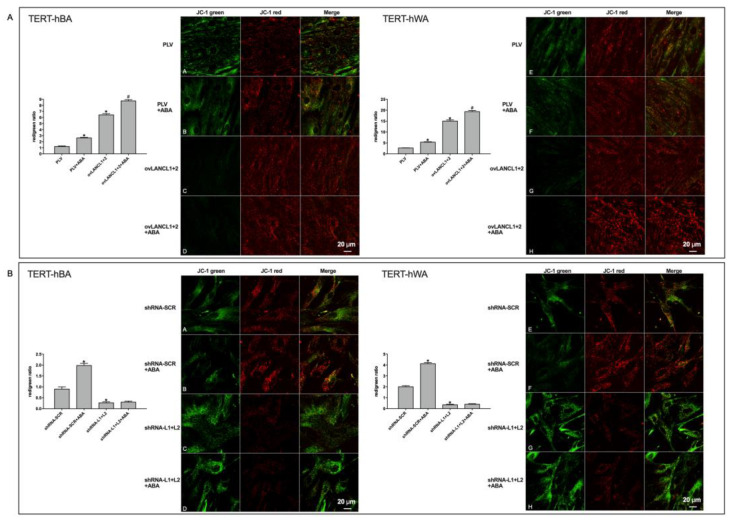
LANCL1/2-expression levels control the mitochondrial proton gradient in human white and brown adipocytes. Fully differentiated TERT-hBA and TERT-hWA cells overexpressing LANCL1 and LANCL2 (ovLANC1+2), or double-silenced for the expression of both proteins (shRNA-L1+L2), and their respective controls, cells infected with the empty vector (PLV) or cells infected with the scrambled silencing sequences (shRNA-SCR), were loaded with the ∆Ψ-sensitive ratiometric fluorescent dye JC-1, and cultured/incubated for 4 h without or with 100 nM ABA. Representative confocal microscopy images of brown (**left panels**) and white (**right panels**) adipocytes overexpressing LANCL1/2 (**A**) or silenced for both proteins (**B**). An increase in red fluorescence indicates a higher ∆Ψ. The histograms show the mean ± SD of the red/green fluorescence ratio calculated in at least 3 microscopic fields for each experiment. * *p* < 0.01 relative to untreated control cells and # *p* < 0.05 relative to ABA-treated control cells by unpaired *t*-test.

**Figure 9 ijms-24-03489-f009:**
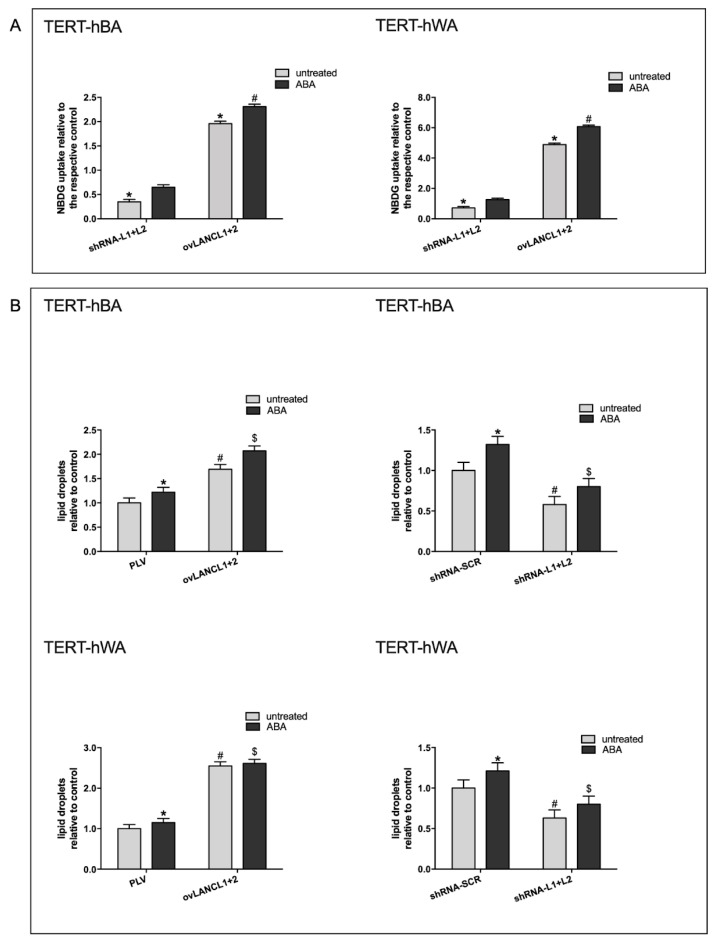
LANCL1/2 expression levels control glucose uptake and affect triglyceride synthesis in white and brown adipocytes. Fully differentiated brown (TERT-hBA) and “beige” TERT-hWA adipocytes overexpressing LANCL1 and LANCL2 (ovLANC1+2), or double-silenced for the expression of both proteins (shRNA-L1+L2), and their respective controls, cells infected with the empty vector (PLV) or cells infected with the scrambled silencing sequences (shRNA-SCR), were analyzed for glucose transport and for triglyceride content. Cells were normalized to total cellular protein content by Bradford assay. (**A**) Glucose transport assays; cells were serum-starved for 12 h, then treated with 100 nM ABA for 5 min prior to incubation with the fluorescent glucose analog 2-NBDG. Results are expressed as NBDG uptake relative to the respective, ABA-untreated control (shRNA-SCR for silencing and PLV for overexpression, not shown in the figure). TERT-hBA, left panel; TERT-hWA, right panel. * *p* < 0.0004 relative to the respective untreated control; # *p* < 0.001 relative to the respective ABA-treated shRNA-SCR or PLV-infected cells. All results are the mean ± SD from at least 3 separate experiments. (**B**) Triglyceride content; triglycerides were stained with Oil Red O and quantitative measures were obtained by spectrophotometric absorbance of isopropanol extracts. TERT-hBA, adipocytes overexpressing LANCL1/2 (upper left panel) or silenced for both proteins (upper right panel); TERT-hWA, adipocytes overexpressing LANCL1/2 (lower left panel) or silenced for both proteins (lower right panel). * *p* < 0.05 relative to untreated control, # *p* < 0.005 relative to untreated control, $ *p* < 0.005 relative to ABA-treated shRNA-SCR or PLV-infected cells. All results are the mean ± SD from at least 4 separate experiments.

**Figure 10 ijms-24-03489-f010:**
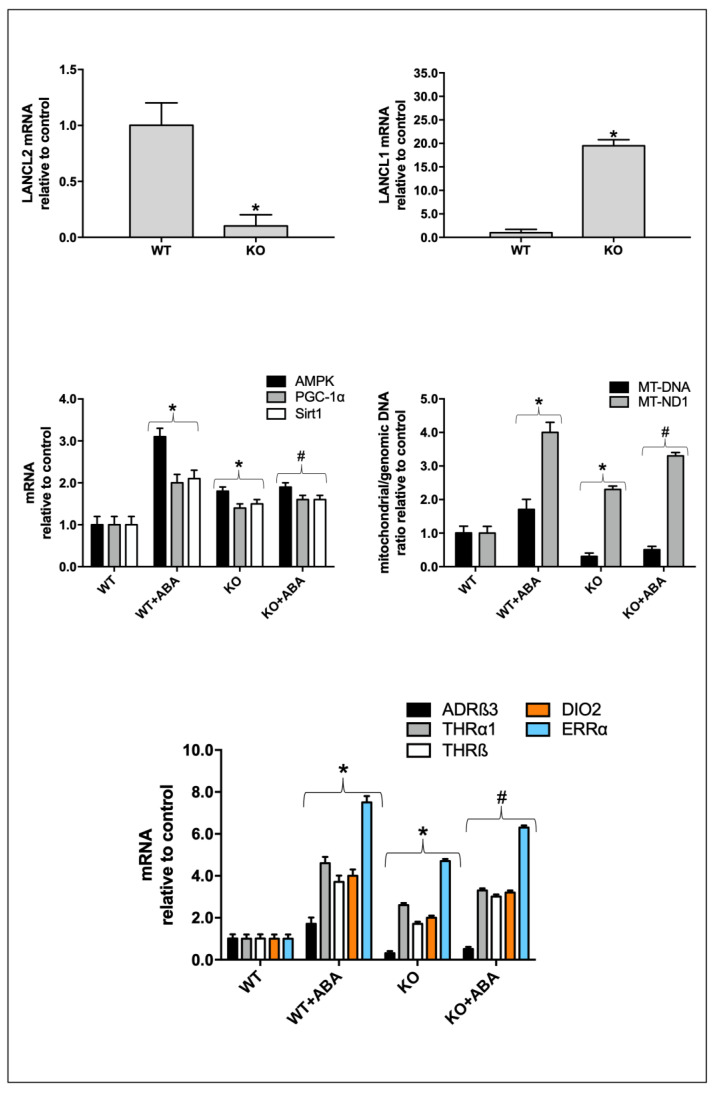
Chronic ABA treatment increases transcription of “browning” receptors, enzymes and transcription factors and mitochondrial DNA content in the BAT of *LANCL2*^−/−^ mice. Male, *LANCL2*^−/−^ mice (KO) and their wild-type siblings (WT), 5 per group, were treated without or with ABA (1 μg/kg BW/day, administered in the drinking water) for 4 weeks. At the end of treatment, mice were euthanized, and samples of brown adipose tissue were taken for qPCR analysis. Results shown are mRNA levels relative to the WT expression level and are the mean ± SD from 5 mice per group. *p* values are calculated by unpaired *t*-test. Upper left panel, LANCL2 mRNA; upper right panel, LANCL1 mRNA; central left panel, AMPK, PGC-1α and Sirt1 mRNAs; central right panel, mitochondrial/genomic DNA ratio (MT-DNA and MT-ND1); lower panel, β-adrenergic receptor 3 (ADRβ3), thyroid hormone receptor α1 (THRα1), thyroid hormone receptor β (THRβ), the enzyme deiodinase 2 (DIO2) and estrogen related receptor alpha (ERRα) mRNAs. * *p* < 0.008 relative to WT; # *p* < 0.05 relative to KO.

**Figure 11 ijms-24-03489-f011:**
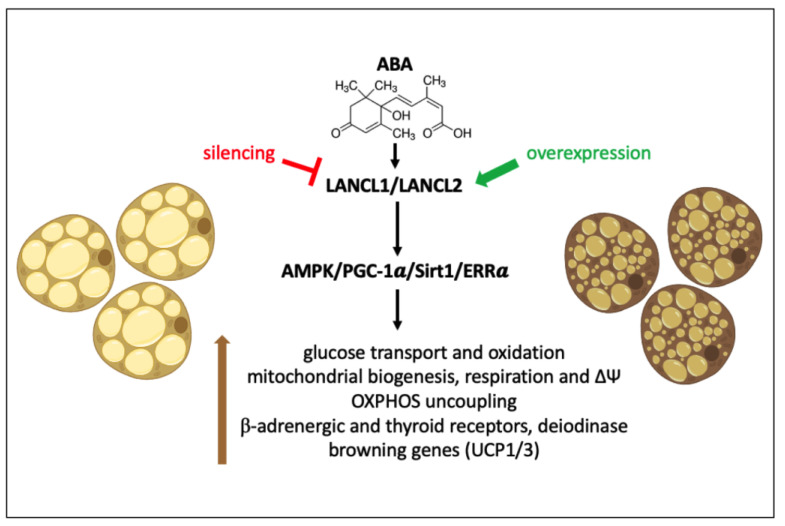
Schematic representation of the role of the ABA/LANCL1/2 hormone/receptor system in the control of energy metabolism, of adipocyte browning and of thermogenesis. Either one of the LANCL proteins can in turn activate the AMPK/PGC-1α/Sirt1/ERRα pathway, increasing glucose transport and oxidation, mitochondrial biogenesis and respiration, OXPHOS uncoupling and T3 and β-adrenergic receptors. The regulatory pathway was probed by overexpressing or silencing the LANCL proteins.

## Data Availability

Not applicable.

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
