# Peer review of "The ABA/LANCL1/2 Hormone/Receptor System Controls Adipocyte Browning and Energy Expenditure"

_ijms, 2023, doi:10.3390/ijms24043489_

Round 1
Reviewer 1 Report
In this manuscript, the authors studied the ABA/LANCL1/2 signaling pathway in human white and brown adipocytes. Overall, the study is very interesting and increase the understanding of this new pathway in adaptive themogenesis.
I was confused with the statistics: in figure 1B (and others), add a line encompassing the 3 genes between the bars and the * or #, otherwise it seems the significance is only for PGC1a and not the other genes. Same for figures 4, 5, and 10.
For figure 6 (mitochondrial respiration), uncoupled respiration should be calculated as the difference between the oligomycin and the rotenone/antimycin A injections. This measure would be more useful than the effect of T3 treatment that is complex to interpretate. In addition, the assays could be performed after norepinephrine or isoproterenol activation to reveal the uncoupling effect of ABA.
The measure of mitochondrial content by qPCR should be a ratio of mitochondrial/genomic DNA primer set. In addition, there is no method section describing the DNA isolation.
In the Lancl2 KO mice, the authors did not measure the AMPK/PGC1a/Sirt1 pathway. Please, explain. In addition, the authors should characterize the browning of white adipose tissues (histology and/or gene expression) as they did in brown fat.
Author Response
1-I was confused with the statistics: in figure 1B (and others), add a line encompassing the 3 genes between the bars and the * or #, otherwise it seems the significance is only for PGC1a and not the other genes. Same for figures 4, 5, and 10.
Brackets have been added over bars in Figures 1, 4, 5 and 10, to identify the genes sharing the same p value with respect to the control.
2-For figure 6 (mitochondrial respiration), uncoupled respiration should be calculated as the difference between the oligomycin and the rotenone/antimycin A injections. This measure would be more useful than the effect of T3 treatment that is complex to interpretate.
The suggested calculations have been performed and results are shown in a new Figure in the Supplementary Data (Figure 2S). Comments regarding these new data have also been added to the Results (page 16, line 497). These data confirm an increased mitochondrial uncoupling in LANCL1/2 overexpressing cells and conversely a reduction in LANCL1/2-double silenced cells, compared with their respective controls. Moreover, these new calculations confirm the results shown in Figure 6, i.e. that the uncoupling effect of ABA on “beige” adipocytes is higher than that of T3.
The reason for showing the effect of T3 was to compare the effect of a known stimulator of mitochondrial uncoupling with that of ABA, a hormone for which an uncoupling effect was hitherto unknown.
In addition, the assays could be performed after norepinephrine or isoproterenol activation to reveal the uncoupling effect of ABA.
The fact that LANCL1/2 overexpression per se increases transcription of beta-adrenergic receptors in brown and white human differentiated adipocytes is expected to increase cell sensitivity to their uncoupling effect, independently of the presence or absence of added ABA.
Indeed, we plan to proceed in our investigation on the role of the ABA/LANCL system in thermogenesis by exploring the interplay between this hormone/receptor system and other hormones (T3 and catecholamines), both in vitro and in vivo. This is just a first report, which aims at providing admittedly preliminary and non-exhaustive evidence, needing further in-depth investigation. These considerations have been added to the Discussion (page 24, line 768).
3-The measure of mitochondrial content by qPCR should be a ratio of mitochondrial/genomic DNA primer set. In addition, there is no method section describing the DNA isolation.
A new paragraph (4.12) has been added to the Materials section (page 29, line 961) to describe DNA isolation and to explain how the mitochondrial/genomic DNA ratio was calculated. In addition, the title of the y-axis of the panels showing the mt/genomic DNA ratio (Figures 1, 4, 5 and 10) has been modified according to the calculation that was in fact applied to obtain the numbers: instead of “fold-change relative to control” it now indicates “mitochondrial/genomic DNA ratio relative to control”.
4-In the Lancl2 KO mice, the authors did not measure the AMPK/PGC1a/Sirt1 pathway. Please, explain.
Transcriptional levels of these genes were explored on cDNA samples from the BAT of WT and Lancl2 KO mice and results have been added to Fig. 10, central panel. A comment has also been added to the Results (page 22, line 664).
In addition, the authors should characterize the browning of white adipose tissues (histology and/or gene expression) as they did in brown fat.
Unfortunately, we did not save any WAT samples from these animals at the time of the study. This is surely a limitation, however, it should be noted that the aim of this study, as stated in the Introduction, was to investigate the role of the ABA/LANCL system in human white and brown adipocyte thermogenesis. We limited the in vivo study on WT and Lancl2 KO mice to analysis of the BAT expression levels of receptors for “browning” hormones, as a cross-species confirmation of a role for the ABA/LANCL system on BAT expression of these receptors seemed to us an important additional piece of information.
We agree that an in-depth investigation of the in vivo effect of targeting the ABA/LANCL system for WAT “browning” purposes is both necessary and informative, however, we do believe that this should be an independent study. This consideration has been added to the Discussion (page 24, line 768).
Reviewer 2 Report
The authors investigate the role of the ABA/LANCL system in the thermogenesis of human white and brown adipocytes. The manuscript is clearly written and well organized. In general, the report is logical with appropriate design however the authors should consider the following points:
- why the authors did not present in the figures the AMPK activity as the p-AMPK/AMPK ratio?
- in materials and methods it is not clear how the authors measured the content of mitochondrial DNA, to be detailed.
- furthermore, the authors do not explain the dose treatment, the choice of cell lines treatment with 100 nM of ABA. should be inserted in materials and methods section.
- It should be discussed the limitations of study
- in materials and methods, in Statistical analysis, the authors indicate the Software, GraphPad Prism Software, but not the type of Statistical analysis and post-test used, to be included in “4.12. Statistical analysis”.
Author Response
- why the authors did not present in the figures the AMPK activity as the p-AMPK/AMPK ratio?
The p-AMPK/AMPK ratios have been added to Fig. 2B and 3B and a new comment has been added to the Results (page 8, line 278).
- in materials and methods it is not clear how the authors measured the content of mitochondrial DNA, to be detailed.
A new paragraph (4.12) has been added to the Materials section (page 29, line 961) to describe DNA isolation and to explain how the mitochondrial/genomic DNA ratio was calculated. In addition, the title of the y-axis of the panels showing the mt/genomic DNA ratio (Figures 1, 4, 5 and 10) has been modified according to the calculation which was in fact applied to obtain the numbers: instead of “fold-change relative to control” it now indicates “mitochondrial/genomic DNA ratio relative to control”.
- furthermore, the authors do not explain the dose treatment, the choice of cell lines treatment with 100 nM of ABA. should be inserted in materials and methods section.
The choice of an ABA concentration of 100 nM for all experiments derives from the following considerations: i) this concentration has been shown to be effective on skeletal myocytes and on cardiomyocytes, stimulating oxidative metabolism, mitochondrial respiration and uncoupling (refs 9 and 16 of the manuscript); ii) normal plasma ABA is in the low nanomolar range in humans (ref 5 of the manuscript) and it increases 20-60 times after dietary intake of ABA-rich fruits or vegetal extracts (ref 6 of the manuscript). Thus, a concentration of 100 nM for ABA was chosen in the present study because it is both effective in vitro on previously explored cell types and also attainable in vivo. These considerations have been added to the Materials and Methods section (paragraph 4.1).
- It should be discussed the limitations of study
A paragraph has been added to the Discussion highlighting the limitations of this study (page 24, line 768).
This is admittedly a preliminary study, undertaken to test the hypothesis that the ABA/LANCL hormone/receptors system could play a role in the regulation of mitochondrial respiration and uncoupling on thermogenic brown and “beige” human adipocytes, similarly to what previously observed on rodent skeletal and cardiac myocytes. Results obtained warrant further preclinical in vivo studies to assess the effect of targeting the ABA/LANCL system on white and brown adipocytes to increase energy expenditure in the adipose tissue, which should reduce body weight gain under high-calories diets. In particular, future studies should focus on the possibility of inducing “browning” of the WAT in vivo, by targeting the ABA/LANCL system, as a means to increase the mass of adipocytes involved in energy expenditure in humans, which possess lower amounts of BAT compared with other mammals.
A dietary intake of ABA, leading to plasma concentrations of the hormone in the high nanomolar, as used in this study, is attainable through increased intake of ABA-rich fruits (ref 12 of the manuscript) or of vegetal extracts (ref 13 of the manuscript), in the form of nutraceuticals.
- in materials and methods, in Statistical analysis, the authors indicate the Software, GraphPad Prism Software, but not the type of Statistical analysis and post-test used, to be included in “4.12. Statistical analysis”.
A new sentence has been added to the paragraph “Statistical analysis” indicating the test used. We apologize for the omission.